# Which public health interventions are effective in reducing morbidity, mortality and health inequalities from infectious diseases amongst children in low- and middle-income countries (LMICs): An umbrella review

Elodie Besnier[1]*, Katie Thomson[2], Donata Stonkute[3]¤, Talal Mohammad[3], Nasima Akhter[4], Adam Todd[5], Magnus Rom Jensen[6], Astrid Kilvik[7], Clare Bambra[2]

1 Department of Sociology and Political Science, Centre for Global Health Inequalities Research (CHAIN), Norwegian University of Science and Technology (NTNU), Trondheim, Norway, 2 Population Health Sciences Institute, Newcastle University, Newcastle upon Tyne, United Kingdom, 3 CHAIN, Department of Public Health and Nursing, NTNU, Trondheim, Norway, 4 Department of Anthropology, Durham University, Durham, United Kingdom, 5 School of Pharmacy, Newcastle University, Newcastle upon Tyne, United Kingdom, 6 Library Section for Humanities, Education and Social Sciences, NTNU, Trondheim, Norway, 7 Medicine and Health Library, NTNU, Trondheim, Norway

☯ These authors contributed equally to this work.
¤ Current address: Max Planck Institute for Demographic Research, Rostock, Germany
* elodie.besnier@ntnu.no

## Abstract

Despite significant progress in the last few decades, infectious diseases remain a major threat to child health in low- and middle-income countries (LMICs)—particularly amongst more disadvantaged groups. It is imperative to understand the best available evidence concerning which public health interventions reduce morbidity, mortality and health inequalities in children aged under five years. To address this gap, we carried out an umbrella review (a systematic reviews of reviews) to identify evidence on the effects of public health interventions (promotion, protection, prevention) on morbidity, mortality and/or health inequalities due to infectious diseases amongst children in LMICs. Ten databases were searched for records published between 2014–2021 alongside a manual search of gray literature. Articles were quality-assessed using the Assessment of Multiple Systematic Reviews tool (AMSTAR 2). A narrative synthesis was conducted. We identified 60 systematic reviews synthesizing 453 individual primary studies. A majority of the reviews reported on preventive interventions (n = 48), with a minority on promotion (n = 17) and almost no reviews covering health protection interventions (n = 2). Effective interventions for improving child health across the whole population, as well as the most disadvantaged included communication, education and social mobilization for specific preventive services or tools, such as immunization or bed nets. For all other interventions, the effects were either unclear, unknown or detrimental, either at the overall population level or regarding health inequalities. We found few reviews reporting health inequalities information and the quality of the evidence base was generally low. Our umbrella review identified some prevention interventions that might

**Data Availability Statement:** This review exclusively worked with information available from published reviews. A list and description of these reviews is available in S9 Appendix.

**Funding:** AT and CB's contributions were supported by the Norwegian Research Council BEDREHELSE work programme (CHAIN: Centre for Global Health Inequalities Research, project number 288638, https://www.forskningsradet.no/en/about-the-research-council/programmes/bedrehelse/). The Norwegian Research Council had no role in study design, data collection and analysis, decision to publish, or preparation of the manuscript.

**Competing interests:** The authors have declared that no competing interests exist.

be useful in reducing under five mortality from infectious diseases in LMICs, particularly amongst the most disadvantaged groups.

## Introduction

The reduction of child mortality and improvement of child health over the last thirty years has been one of the greatest successes achieved in global health [1]. Globally, under five mortality has declined from 85.9 deaths per 1 000 live births in 1990 to 37.1 in 2019 [2–5]. However, this burden remains unevenly distributed both within and between countries. According to the Institute for Health Metrics and Evaluation (IHME) Global Burden of Diseases (GBD) estimates for 2019, under-5 mortality represented more than a third of all deaths in low-income countries whereas it is less than 10% of all deaths globally [5]. Within countries, inequalities in health have been identified across many socio-economic factors. Household surveys from the Millennium Development Goals era show that children from the poorest households are almost twice as likely to die before the age of five compared to their wealthiest counterparts [1]. Similarly, children born from mothers with no education are almost three times as likely to die than those born from mothers with secondary or higher education [1]. Inequalities are not only found between the most privileged and the most deprived groups but also *within* deprived groups. For example, a 2018 report by the United Nations Children's Fund (UNICEF) reveals that inequalities in child health indicators and outcomes not only vary between rural and urban populations but also within urban populations [6].

Despite major progress in fighting infectious diseases [7, 8], the disease burden in children under five remains significant, especially in low- and middle-income countries (LMICs) [9]. According to IHME 2019 GBD estimates, lower respiratory infections such as pneumonia, diarrheal diseases, malaria, HIV/AIDS and even vaccine-preventable diseases like measles remain among the leading causes of deaths and illness in children under five living in LMICs [9]. Since effective means of prevention and control for many infectious diseases exist [10–12], any inequalities in the burden of these diseases between population groups or countries are an equity issue as they are "avoidable inequalities in health between groups of people within countries and between countries [arising] from inequalities within and between societies" [13].

The evidence base concerning the effect of public health interventions on health inequalities in children is growing globally. However, gaps remain. Among these gaps in the literature on child health and public health interventions is a persistent lack of explicit or broad focus on equity issues in systematic reviews [11, 14–17]. Additionally, the majority of available evidence on equity and public health comes from high-income countries, for children and adults alike [14, 18–20].

In this umbrella review, we aim to address some of these gaps by searching for public health interventions that are effective in reducing morbidity, mortality and health inequalities from infectious diseases (as defined by the 11th International Classification of Diseases [21]) amongst children under five years of age living in LMICs. More specifically, we aim to answer the following research questions:

- Which public health interventions are effective in reducing morbidity and mortality from infectious diseases amongst children in LMICs?

- What are the effects of these interventions on health inequalities?

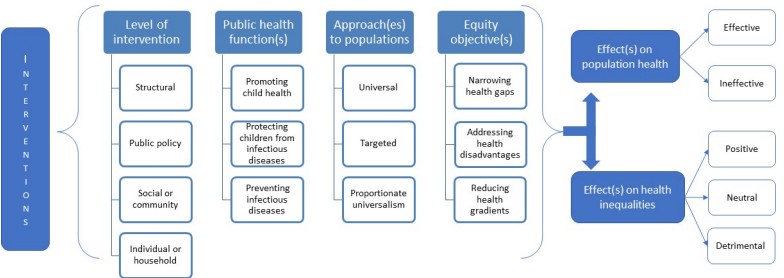

**Fig 1. Analytical framework to understand the effect of public health interventions on health inequalities in LMICs [33].**

## Model, framework or theory

In line with the concept of population health [22], we defined public health interventions as policies, programs or actions aiming at "preventing disease, prolonging life and promoting health through the organized efforts of society" [23] and "shift[ing] the distribution of health risk by addressing the underlying social, economic and environmental conditions" [24].

The analytical framework of our review (Fig 1) builds on the framework proposed by Bambra et al. [25], itself developed from the health inequalities conceptual literature [26–28]. Using this framework, we identified four levels of interventions: *the structural and macro-policy level* (the macro-economic, cultural and environmental context that influences the living standards of the whole population), *the public policy level* (policies that influence the environments in which people live, work or study), *the social networks and community level* (the collective actions that affect the health of communities and local areas by building social cohesion and mutual support), and *the individual or household level* (the interventions and strategies targeting the health of individual people or households). Then, we identified three conceptual intervention approaches to populations and health inequalities, as well as three equity objectives for these interventions. These include *targeted approaches* (directed at specific groups—i.e. deprived, vulnerable or disadvantaged groups—in a population), *universal approaches* (interventions that apply uniformly to the population) or *proportionate universalism* (interventions that are applied universally but their intensity and scale is proportionate to the level of disadvantage or health gradient across that population) [29]. Following Graham's typology [26], our framework identifies three equity objectives for these interventions: "*remedying health disadvantages*", which addresses the health needs of the most deprived or disadvantaged population; "*narrowing health gaps*", which focuses on reducing the difference in health found between the most privileged and the most disadvantaged groups; and "*reducing health gradients*", which aims to reduce health differences across the whole population.

To reflect this umbrella review's focus on public health interventions, our framework was augmented with the three core public health functions or services identified by the World Health Organization Regional office for Europe's (WHO EURO) as essential public health operations [30]: *promoting* child health, *protecting* children from infectious diseases and *preventing* such diseases. Following WHO EURO's definition, health promotion is defined as intersectoral and interdisciplinary operations enabling people to maintain or improve their health and its determinants. Health protection covers the use of legal, regulatory or enforcement mechanisms to safeguard public health. Finally, prevention involves the targeting of individuals or populations at risk of developing a disease using public health services within the

health system [30]. However, WHO EURO recognizes that certain actions may overlap between these three different public health functions [30]. Therefore, we used the Campbell Collaboration and UNICEF Office of Research- Innocenti evidence mega-map on child welfare [15] and feedback from researchers contacted by the review team to identify and define broad types of interventions under each of the three core public health functions (see S1 Appendix). Secondary prevention activities (early detection of a disease before it becomes symptomatic), tertiary prevention activities (involving improving treatment and recovery, improving the health outcomes in those already affected by a disease) [31] and curative approaches were excluded.

## Materials and methods

We conducted an umbrella review to identify systematic reviews or evidence synthesis of public health interventions which reduce morbidity, mortality and/or health inequalities due to infectious diseases amongst children (aged under five years) in LMICs. An umbrella review, also called an 'overview of reviews' or 'review of reviews', involves the compilation and synthesis of evidence from multiple (systematic) reviews into a single, easy-to-use document [32]. The full methodology has been described in a published protocol (see S2 Appendix) [33], which was also registered with PROSPERO (Registration number: CRD42019141673). The PRISMA-E checklist [34] for this review is attached in S3 Appendix.

### Search strategy

The following databases were searched from January 2000 until January 2021 (by EB): the Cochrane Library (includes the Cochrane Database of Systematic Reviews, the Cochrane Central Register of Controlled Trials and the Cochrane Clinical Answers), Medline (Ovid), EMBASE (Ovid), the CAB Global Health database (Ovid), Health Evidence (McMaster University), the Campbell Collaboration Library of Systematic Reviews (The Campbell Library), International Initiative for Impact Evaluation Systematic review repository (International Initiative for Impact Evaluation - 3ie), Scopus (Scopus), the Social Sciences Citation Index (SSCI, Web of Science) and Prospero (Centre for Reviews and Dissemination, University of York). These search dates allowed us to capture the increased efforts in improving child health further to the adoption of the Millennium Development Goals (MDGs). However, given the number of reviews captured in the search that had been updated several times within these dates, the records were later limited to records published from January 2014 (i.e. in the five years before the initial search) until January 2021 (the date of the updated search). Research librarians (MRJ, AK) provided guidance and support in the choice of databases and the design of the search strings. These search strings involved a combination of MeSH terms and free-text keywords. They were piloted in Medline (via Ovid) and Scopus (see S4 Appendix). Once finalized, we (EB) adapted the search strings for each database. To facilitate their translation from one database to the other, we used the Polyglot Search Syntax Translator [35, 36], No restriction of language was applied.

In addition to the searching of these databases, we (EB) performed a manual search in Google Scholar and on the following international organizations' websites, using selected keywords:

- UNICEF Office of Research–Innocenti [37]

- UNICEF [38, 39]

- World Health Organization (WHO) [40, 41]

The full search strategy for both the databases and manual searches can be found in S5 Appendix.

Finally, and in addition to what was stated in our protocol, we (EB/TM or EB/DS) conducted citation follow-up by searching the reference lists of umbrella reviews captured by this search for potential additional records. The umbrella reviews themselves were excluded from our review. For the systematic review protocols captured in our search, we (EB, TM, DS) searched for the review's full text or contacted the authors to inquire about the status of the review.

## Selection criteria

The inclusion/exclusion criteria (Table 1) were defined *a priori* using PICOS:

- **Population**: children under five years old or households with at least one child under five years old, living in countries that have been listed as low-, lower middle or upper middle-income by the World Bank at least once from 2000 to present [42].

- **Intervention**: public health interventions targeting infectious diseases or associated risk factors in children, as defined in our framework.

- **Comparison/control**: systematic reviews and evidence synthesis of primary studies with control groups or other comparison groups, such as pre- and post- or alternative intervention comparisons. If no control or comparison was provided, the study was excluded.

**Table 1. Inclusion/exclusion criteria.**

| Inclusion | Exclusion |
|---|---|
| The review team has access to the publication's full text. | The publication's full text cannot be obtained by the review team (e.g. reviews withdrawn or unpublished) |
| The article was published after 2014. | The article was published before 2014. |
| The publication is an academic article or a report of a systematic review (including a meta-analysis) or an evidence synthesis as defined in PICOS, synthesizing at least 2 relevant primary studies. | The publication is a primary study, an umbrella review, a conference proceeding or paper, an abstract, editorial, letter, comment, erratum, survey, note or a doctoral thesis; or does not meet one or more of the three key elements of systematic reviews and evidence syntheses as defined in PICOS; or does not synthesize at least 2 relevant primary studies. |
| The publication covers exclusively or synthesizes separately, studies in countries defined at least once since 2000 as low-, lower-middle or upper-middle-income by the World Bank historical classification [42]. | The publication only includes interventions in country/ies the World Bank historical classification has continuously defined as high-income between 2000 and 2019 [42], or does not synthesize or report on low-, lower-middle or upper-middle-income countries separately. |
| The publication covers interventions targeting children from livebirth until five years old or households with children under five years old. | The publication only includes interventions targeting adults, pregnant women, adolescents or children older than five years old; or fail to synthesize primary studies' results for the under-5 age group separately. |
| The publication covers active, collective health promotion, health protection or primary prevention public health interventions addressing or affecting the burden of infectious diseases or their risk factors. | The publication only includes curative interventions or secondary or tertiary prevention interventions; is not addressing or affecting the burden of infectious diseases or their risk factors; or only report trends in individual behaviors without any actions aiming at changing or influencing them. |
| The publication reports health or health inequalities outcomes in and between populations, disaggregated by one or more of the PROGRESS+ factors as defined in PICOS. | The publication does not include a relevant overall health outcome or disaggregated information by or between population groups. |

- **Outcomes**: both health and health inequality outcomes reflecting the effectiveness of the intervention. Primary outcomes included overall population level measures of mortality, morbidity or service uptake and coverage reflecting the effectiveness of the intervention. Secondary outcomes included health inequality measures (defined as variations between groups or populations) in these primary health outcomes according to the Progress + factors [43, 44]: Place of Residence, Race/ ethnicity/ cultural background, Occupation, Gender and sex, Religion, Education, Social capital, Socio-economic status, Others (e.g. age or health status)

- **Study design**: systematic reviews (including meta-analyses) and evidence syntheses covering at least two relevant primary studies and published since 2014. The primary studies covered by individual reviews included both randomized and non-randomized design. Following the criteria of the Database of Abstracts of Reviews of Effects (DARE) [45], we identified three key elements for systematic reviews or evidence syntheses to be included in this umbrella review: 1) a clear question, 2) a transparent method for the search, selection and appraisal of evidence or studies and 3) a separate synthesis of the results or evidence meeting this umbrella review's scope and inclusion criteria. When a review had been updated, only the most recent version was included.

## Screening

We used Endnote x9 to remove duplicates from the list of citations [46]. The duplicates that failed to be captured by the software were removed manually during screening. Article screening was carried out using the software Rayyan [47]. For articles in languages other than English, reviewers were supported by a translator or a native speaker. One reviewer (EB) screened citation titles and abstracts. A random ten percent sample was screened independently by a second reviewer (KT). Discrepancies were resolved by consensus and with the discussion with the team (DS, TM, AT). Agreement between reviewers was high (95% of the articles, kappa score $K$ = 0.63). All full texts were assessed independently by two reviewers (first team: TM, EB; second team: DS, EB). Discrepancies were resolved by consensus or with the arbitration of the third reviewer. Agreement was high (85% of the articles, kappa score $K$ = 0.63 for the first team, 94% of articles, kappa score $K$ = 0.65 for the second team).

## Data extraction

The extraction template was developed a priori at the same time as the protocol for this review. This form (S6 Appendix) was piloted by three reviewers (EB, TM, DS) and checked by a fourth (KT) using a sample of three articles [48–50]. Once the template was finalized, individual articles were extracted by one reviewer and checked by a second.

## Quality appraisal

Each extractor (EB, TM, DS) critically appraised individual articles using the Assessment of Multiple Systematic Reviews tool (AMSTAR2) at the same time as the data extraction (see S6 Appendix) [51]. Then, this appraisal was checked by a second reviewer (EB, TM, DS). Discrepancies were resolved by consensus.

## Overlaps between studies

As part of the extraction, each reviewer listed the relevant primary studies covered by individual review into a citation matrix developed by Thomson et al. [52] in order to identify overlaps

(S6 Appendix). This list was checked by a second reviewer alongside the critical appraisal and extraction sheet.

## Data synthesis

The broad scope of interventions, study designs and type of outcomes included in this umbrella review made quantitative analysis difficult. Hence, the systematic reviews were narratively synthesized using the framework in Fig 1. The results were grouped first by level of interventions, public health functions, and then by approach to population for analysis. To ensure the comparability of individual reviews' results, they were synthesized according to broad types of interventions previously defined (S1 Appendix). When contradictory or heterogeneous results were found within the same category, these were explored according to the quality of the review, the quality of the evidence base, the characteristics of the review. the details of the intervention and the detail of the population and setting of the intervention. The findings of high- and moderate-quality reviews were also synthesized separately for each category. When narrating findings from reviews synthesizing together different levels of intervention, we took the pragmatic decision to synthesize together the results for the structural and policy levels, and those for the community, households and individual levels. Due to the number of reviews covering certain types of interventions, these were further divided into smaller categories according to their aim or main components.

## Ethical approval

This review exclusively worked with anonymous, group-level information available from published reviews. As a result, there is no risk to identifying individual data or disclosing confidential information. This study did not require ethical approval.

## Results

As shown in the PRISMA chart (Fig 2), the database searches identified 17 895 citations while the website searches identified 105 records. After removing duplicates, a total of 8 980 unique citations were screened for titles and abstracts, leading to 393 full texts being assessed. The reference lists of umbrella reviews captured by these searches were screened manually but did not identify any further citations matching our criteria that were not already captured by previous searches. Finally, 60 systematic reviews reporting on 453 individual primary studies (587 references) were included in our qualitative synthesis. The list of excluded records at full text assessment and reasons for exclusions can be found in S7 Appendix.

Of the 453 individual primary studies covered, twenty-one percent were covered in more than one review (S8 Appendix). For each broad type of intervention, we identified the number of studies overlapping across reviews using the citation matrix developed by Thomson et al. [52]. We reported the number of individual studies actually covered in the results, to reflect the size of the evidence base. Our umbrella review focused on reviews' syntheses and did not re-analyze the findings of the primary studies covered. Hence, we did not exclude overlapping studies from our synthesis. However, these overlaps were reflected in our analysis of the evidence base and heterogeneity of findings on individual intervention's impact.

## Quality of the evidence

Overall, the quality of the reviews was mixed (see Table 2). While 57% of them rating as low or critically low on the AMSTAR 2 tool, a third were rated as high quality and 10% were of moderate quality. Fig 3 shows the occurrence of methodological and reporting weaknesses from the AMSTAR 2 checklist found across the 60 reviews we included.

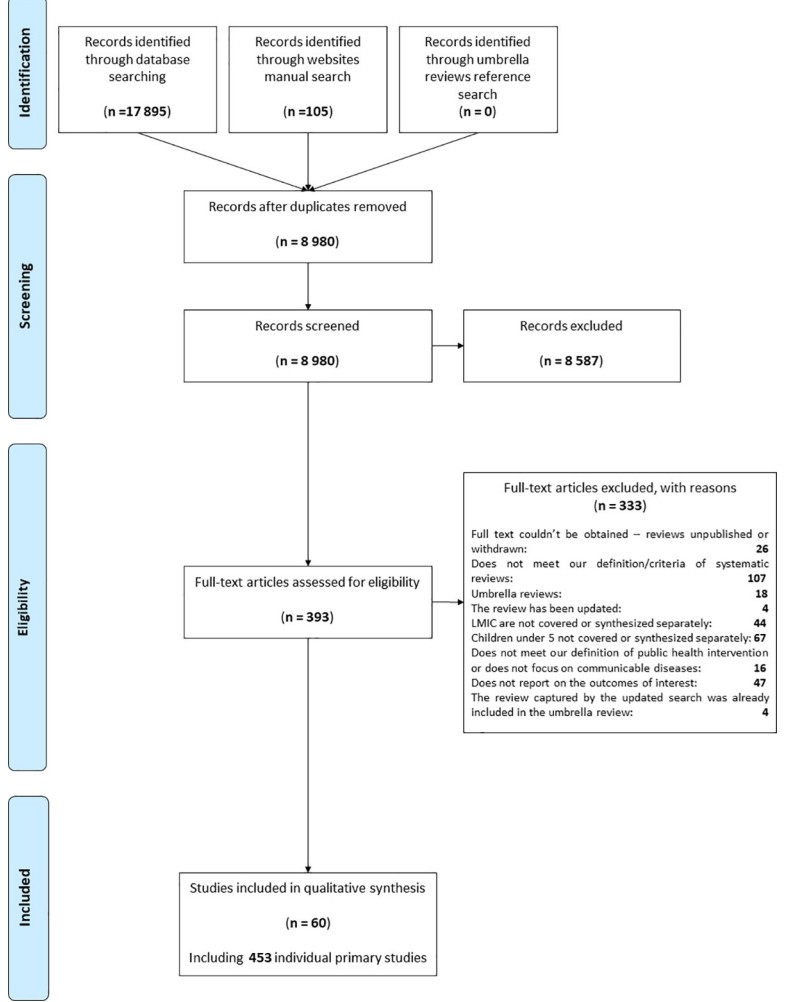

**Fig 2. PRISMA chart** [53].

## Summary of studies characteristics

Included reviews covered 23 narrative or qualitative reviews, 20 meta-analyses and 17 mixed-methods reviews which had both a quantitative and qualitative synthesis. The primary studies covered 72 different LMICs with four countries–India, Bangladesh, Brazil and Pakistan–accounting for over a third of the studies (Fig 4).

The vast majority of included reviews covered interventions aiming at preventing infectious diseases in children [49, 50, 54–59, 62, 64, 65, 68, 69, 71–82, 84–99, 101, 102, 104, 106, 108–110]. Over a quarter included health promoting interventions [48, 49, 60–64, 66, 67, 70, 72, 78,

**Table 2. Quality assessment of included reviews.**

| Amstar 2 overall rating | Number of reviews | Reference |
|---|---|---|
| **High** | 20 | [48, 49, 54–71] |
| **Moderate** | 6 | [72–77] |
| **Low** | 9 | [50, 78–85] |
| **Critically low** | 25 | [86–110] |

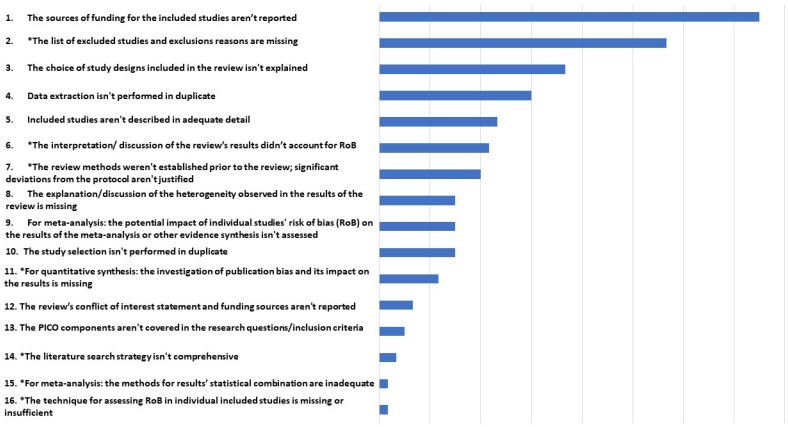

**Fig 3. Included reviews' methodological weaknesses identified according to the AMSTAR2 checklist [51].** (*) Items identified as critical in the AMSTAR2 tool checklist.

83, 92, 103, 107] and only two covered health protection interventions [100, 105]. The description of the various interventions covered can be found in Tables 3–5.

Looking at the number of included reviews per intervention level, two reviews included interventions at the structural level [49, 85], twenty-five covered interventions at the public policy level [49, 55, 58, 62, 64, 68, 69, 72, 78, 81, 83, 85–88, 90, 91, 93, 97, 98, 101, 102, 104, 108, 110], twenty-four at the community level [50, 59, 62, 64, 65, 69, 72–74, 76, 78, 80, 82, 84, 85, 89, 92, 95–97, 99, 100, 106, 109] and twenty-five at the individual level [48, 55–57, 60, 61, 63, 64, 66, 67, 69–71, 75, 77, 79, 80, 84, 90, 92, 94, 99, 103, 105, 107]. As the numbers show, individual reviews often included interventions covering several levels. Hence, we took the pragmatic decision to report together the impact of structural and policy level interventions (found in 25 reviews), and also combined the impact of community, households and individual level interventions (found in 44 reviews), under each of our three key public health functions: promotion, prevention, protection. Four reviews also included multilevel interventions that were analyzed separately [62, 69, 85, 97].

Regarding approaches to health inequalities adopted by the interventions covered, the vast majority of reviews (forty-five in total) included interventions with a universal approach

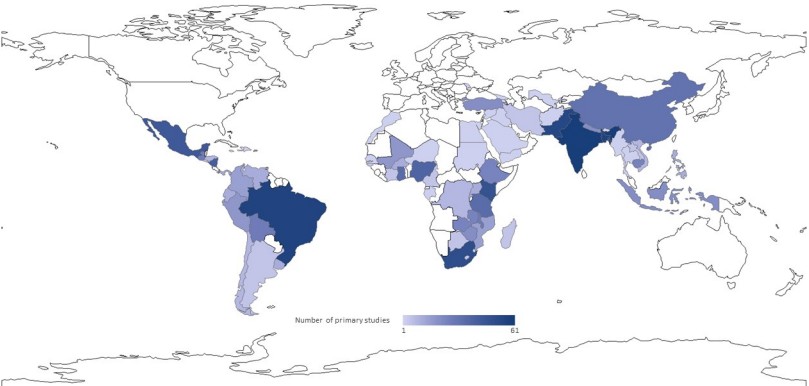

**Fig 4. Distribution of the primary studies covered in included reviews, by countries\*.** * Seven primary studies from Lamberti et al. [93] were only identified by their country development category and world region. These studies couldn't be represented in this map.

**Table 3. Overview of interventions promoting child health covered by the reviews and their impacts on child health and health inequalities.**

| Public health function | Level of intervention | Approaches to population and health inequalities | Intervention (n = number of reviews) | Description of the intervention covered | Impacts on child health at population level | Impacts on health inequalities in children |
|---|---|---|---|---|---|---|
| Promoting child health | Structural and policy levels | Universal | *No review found* | | | |
| | | Targeted | Contracting out health promotion services (n = 1) | Contracting out health promotion services to non-governmental actors to improve service utilization and health outcomes in rural, under-resourced communities. (n = 1) | No effect on diarrhea incidence (n = 1) | Neutral effect on health disadvantages (n = 1) |
| | | | Financial and non-financial assistance and incentives (n = 6) | Non-monetary incentives/assistance (e.g. food distribution) delivered to poor or vulnerable and incentives families and communities at or conditionally to attending certain child health services. (n = 3) | Can improve child immunization coverage (n = 3) | Can have a beneficial effect on health disadvantages (n = 3) |
| | | | | Monetary incentives (e.g. cash transfers or vouchers, with or without additional services) delivered to the poor or vulnerable and incentives to families and communities to attend (or conditionally attend) certain child health services. (n = 5) | Reduces morbidity risks from diarrhea and acute respiratory infections (n = 1) — Has mixed effect on child immunization (n = 5) | Has mixed effect on health disadvantages (n = 5) |
| | Multilevel intervention | Universal | *see Table 5 - Preventing infectious diseases in children (mixed intervention)* | | | |
| | | Targeted | *No review found* | | | |
| | Community, households and individual levels | Universal | Nutrition supplements (n = 6) | Zinc supplementation given weekly or daily to children, with or without other nutrition interventions (n = 3) | Has mixed results on lower respiratory tract infection morbidity (n = 2) — Reduced diarrhea morbidity (n = 1) — No effect of the supplements on otitis morbidity (n = 1) | Insufficient evidence on addressing health disadvantages (n = 2) |
| | | | | Vitamin A to neonates, infants or children alongside other vitamin supplements or selected vaccines. (n = 3) | No effect on morbidity nor mortality from diarrheal or acute respiratory tract infections (n = 3) | *Unknown* |
| | | | | Multiple micronutrient supplementation to healthy children (n = 1) | No effect on diarrheal morbidity (n = 1) | *Unknown* |
| | | | | Iron or Iron/Folic Acid supplementation (n = 1) | No effect on diarrheal morbidity (n = 1) | *Unknown* |
| | | | | Additional animal-source foods provided to children (n = 1) | Has mixed results on diarrhea, respiratory infections or malaria (n = 1) | *Unknown* |
| | | Mixed targeted/universal | Nutrition supplements (n = 4) | Oral probiotics/synbiotics supplementation mixed to milk for neonates (n = 1) | Reduces morbidity from neonatal sepsis (n = 1) | *Unknown* |
| | | | | Multiple micronutrient powders given to caregivers to be mixed with infants and children's food before consumption (n = 3) | Has mixed effect on diarrhea (n = 3), respiratory infections or malaria (n = 1) | Has mixed effect on health disadvantages (n = 2) |
| | | Universal | Health promotion education to caregivers (n = 3) | Health promotion education to caregivers of children under five, carried out by health workers at community or individual levels (n = 3) | Shows mixed results on immunization uptake and coverage (n = 2) — Positive effect on morbidity from diarrhea and malaria. (n = 1) | *Unknown* |

(n =): Number of reviews covering this type of intervention.

Green cell: Beneficial intervention effect on the outcome of interest.

Blue cell: Neutral intervention effect on the outcome of interest.

Orange cell: Inconclusive/mixed intervention effect on the outcome of interest.

Red cell: Detrimental intervention effect on the outcome of interest.

**Table 4. Overview of interventions protecting children from infectious diseases covered by the reviews and their impacts on child health and health inequalities.**

| Public health function | Level of intervention | Approaches to population and health inequalities | Intervention (n = number of reviews) | Description of the intervention covered | Impacts on child health at population level | Impacts on health inequalities in children |
|---|---|---|---|---|---|---|
| **Protecting children from infectious diseases** | Structural and policy levels | Universal | *No review found* | | | |
| | | Targeted | *No review found* | | | |
| | Multilevel intervention | Universal | *No review found* | | | |
| | | Targeted | *No review found* | | | |
| | Community, households and individual levels | Universal | Improved cookstoves (n = 2) | Providing improved cookstoves to households to reduce household air pollution (n = 2) | No effect on child morbidity from pneumonia or acute respiratory tract infections of any severity (n = 2) | *Unknown* |
| | | Targeted | *No review found* | | | |

(n =): Number of reviews covering this type of intervention.

Green cell: Beneficial intervention effect on the outcome of interest.

Blue cell: Neutral intervention effect on the outcome of interest.

Orange cell: Inconclusive/mixed intervention effect on the outcome of interest.

Red cell: Detrimental intervention effect on the outcome of interest.

[48, 50, 55, 58–63, 66–68, 71–74, 77–81, 85–88, 90–102, 104–108, 110]. Twenty-six reviews included interventions adopting a targeted approach [48, 49, 54, 56, 57, 62–65, 67, 68, 70, 72, 75, 76, 78, 82–85, 89, 92, 97, 99, 103, 109], and ten included a mix of both approaches [48, 62, 63, 68, 72, 78, 85, 92, 97, 99]. None of the included reviews described interventions adopting a proportionate universalism approach.

Only twenty reviews reported intervention effects across different groups [48, 50, 58, 63, 68, 69, 72, 77, 80, 81, 83, 88, 93, 94, 97, 99, 102, 104, 109, 110]. Another twenty one reviews reported the interventions' effect on targeted disadvantaged groups as one population [49, 54, 56, 57, 61–64, 67, 75, 76, 82–85, 88, 89, 92, 97, 99, 103]. In terms of factors of disadvantages or variations in health covered by these studies, the most common ones found in the reviews are health-related vulnerabilities or higher needs (e.g. areas with low service coverage, HIV affected families, disease-endemic areas), the place of residence (and the rural/urban divide in particular), and the children's age group (e.g. infant versus children between 1 and 5 years old). Factors related to socio-economic status, parental education or race were far less common.

Tables 3–5 summarize the results from the reviews on the effects of the promotion, prevention and protection interventions on child health and health inequalities. A description of all included reviews is available in S9 Appendix.

## Promoting child health

We found seventeen reviews that evaluated interventions promoting child health [48, 49, 60–64, 66, 67, 70, 72, 78, 83, 85, 92, 103, 107]. Seven covered interventions at the structural and policy intervention level [49, 62, 64, 72, 78, 83, 85], one included multilevel interventions and twelve looked at interventions at the community, households and individual level. [48, 60–63, 66, 67, 70, 78, 92, 103, 107], as summarized in Table 3.

**Promoting child health—Structural and policy level interventions.** Of the twenty-five reviews that cover interventions at the structural and policy levels, seven looked at interventions promoting child health. These covered two main types of interventions: contracting out health promotion services to non-governmental service providers, and financial and non-

**Table 5. Overview of interventions preventing infectious diseases in children covered by the reviews and their impacts on child health and health inequalities.**

| Public health function | Level of intervention | Approaches to population and health inequalities | Intervention (n = number of reviews) | Description of the intervention covered | Impacts on child health at population level | Impacts on health inequalities in children |
|---|---|---|---|---|---|---|
| Preventing infectious diseases in children | Structural and policy levels | Universal | Water supply infrastructure (n = 2) | Improving water quality by improving water supply infrastructure (n = 2) | Insufficient evidence on intervention impact on childhood diarrhea morbidity (n = 2) | Unknown |
| | | | Introduction of new vaccines (n = 14) | Introducing Haemophilus influenzae type b vaccine into national child immunization programs, with or without catch-up doses for children over the vaccination age (n = 2) | Reduces Hib meningitis mortality or morbidity in infants (n = 2) | Unknown |
| | | | | Introducing pneumococcal conjugate vaccines into national child immunization programs, with or without catch-up doses for children over the vaccination age (n = 6) | Reduces mortality or morbidity from pneumococcal infections in children (n = 6) | Reduce the gap between age groups in healthy children in the burden of pneumococcal diseases (n = 3) |
| | | | | | | Risk of increased gap between HIV positive and HIV uninfected children in the burden of pneumococcal diseases (n = 1) |
| | | | | Introducing rotavirus vaccines into national child immunization programs, with or without catch-up doses for children over the vaccination age (n = 8) | Reduces child mortality or morbidity due to diarrhea or gastroenteritis (n = 8) | Reduce the gap between age groups in the burden of gastroenteritis (n = 3) |
| | | | | | | Insufficient evidence on the effect on global health inequalities (n = 3) Has mixed results on the effect in selected disadvantaged groups (n = 1) |
| | | Targeted | Contracting out preventive services (n = 1) | Contracting out preventive health services to non-governmental actors to improve service utilization and health outcomes in rural, under-resourced communities. (n = 1) | No effect on immunization uptake (n = 1) | Neutral effect on health disadvantages (n = 1) |
| | Multilevel intervention | Universal | Multilevel interventions to improve child immunization (n = 3) | Combined demand-and supply-side interventions across levels to improve child immunization (n = 3) | Improves child immunization coverage and/or uptake (n = 3) | Unknown |
| | | | Multilevel Water, Sanitation and Hygiene interventions (n = 1) | Combined hygiene education, water quality and water supply/sanitation infrastructure provision (n = 1) | Reduces diarrhea and dysentery morbidity | Unknown |
| | | Targeted | No review found | | | |
| | Community, households and individual levels | Universal | Increasing child immunization (n = 12) | Increase demand for child immunization through immunization education, communication and social mobilization with or without outreach services (n = 10) | Positive effect on the coverage or uptake of one or more vaccines (n = 10) | Beneficial effect on health disadvantages (n = 3), except for mass communication promoting immunization (n = 2) |
| | | | | Increase the supply of child immunization services through service integration, training of healthcare professionals, introduction of new technologies and improving community or outreach services (n = 5) | Mixed results on immunization uptake and coverage (n = 5) | Risk of increased rural/urban disparities with services integration (n = 2) |
| | | | Increasing bed net use (n = 1) | Mass media campaigns to promote bed net use (n = 1) | Increases bed net use for children (n = 1) | Unknown |
| | | | Water, Sanitation and Hygiene interventions (n = 7) | Water quality improvement at point-of-use at the community or household level (n = 3) | Reduces diarrhea morbidity (n = 3) | Insufficient evidence on the effect on global health inequalities in the diarrhea burden (n = 1) |
| | | | | Sanitation and hygiene intervention at the community or household level (n = 5) | Mixed results on diarrhea morbidity (n = 4) | Insufficient evidence on their effect on the health gap between age groups (n = 1) or settings (n = 2) |
| | | | | | Mixed results on respiratory infections and pneumonia morbidity (n = 2) | |
| | | | | | Mixed results on parasitic diseases morbidity (n = 3) | |
| | | | | | No effect on dysentery prevalence (n = 1) | |
| | | | | | Reduces trachoma morbidity (n = 1) | |
| | | Targeted | Increasing child immunization (n = 7) | Increasing demand and utilization of immunization services, including education sessions, communication and information campaigns, reminder tools with or without outreach services (n = 6) | Improves immunization coverage or uptake (n = 6) | Beneficial effect on health disadvantages (n = 6) |
| | | | | Increase the supply of child immunization services through integration of immunization with other health services, the provision of tools and material to support immunization health professionals or health professional training, either on their own or as part of multi-component interventions (n = 5) | Mixed results on immunization uptake and coverage (n = 5) | Insufficient evidence on addressing health disadvantages (n = 5) |
| | | | | Community outreach education activities to mothers and households with children under five on the use of insecticide treated nets (n = 2) | Increases under-5 use of insecticide treated nets (n = 2) | Beneficial effect on health disadvantage faced by communities in malaria-endemic areas (n = 2) |
| | | | Increasing bed net use (n = 2) | Extended infant HIV prophylaxis during breastfeeding (n = 1) | Lower risks of HIV infection (n = 1) | Beneficial effect on the health disadvantage faced by infants born to HIV positive, breastfeeding mothers (n = 1) |
| | | | Preventing HIV transmission (n = 3) | Home visits by community health workers for HIV exposed infants (n = 2) | Increases uptake of HIV prophylaxis in infants born to HIV positive mothers | Beneficial effect on the health disadvantage faced by infants born to HIV positive mothers |
| | | | | HIV services integration into maternal and child health services (n = 2) | Mixed results on the uptake of HIV prophylaxis in infants born to HIV positive mothers (n = 2) | Mixed results on the impacts on addressing the health disadvantage faced by infants born to HIV positive mothers (n = 2) |
| | | | | Peer-to-peer education, task-shifting and service quality improvement to HIV positive mothers (n = 1) | No effect on HIV prophylaxis uptake in infants born to HIV positive mothers (n = 1) | No effect on addressing the health disadvantage faced by infants born to HIV positive mothers (n = 1) |
| | | | Preventing infections in vulnerable neonates (n = 2) | Kangaroo mother care in hospital settings for low-birth-weight/premature neonates (n = 1) | Reduces the risks of severe illness, nosocomial infection/sepsis and lower respiratory tract diseases in babies (n = 1) | Neutral effect on health disadvantages (n = 1) |
| | | | | | No effect on mild/moderate infection and illness or diarrhea in babies (n = 1) | |
| | | | | Applying topical of emollients (ointments, creams, or oils) to prevent infections in premature infants (n = 1) | No effect on the incidence of invasive infections (n = 1) | Neutral effect on health disadvantages (n = 1) |

(n =): Number of reviews covering this type of intervention.

Green cell: Beneficial intervention effect on the outcome of interest.

Blue cell: Neutral intervention effect on the outcome of interest.

Orange cell: Inconclusive/mixed intervention effect on the outcome of interest.

Red cell: Detrimental intervention effect on the outcome of interest.

financial assistance and incentives. All of these interventions had a targeted approach, focusing on the impacts on the most disadvantaged in society.

*Contracting out health promotion services (1 review).* One high-quality review covering two primary studies explored the effects of contracting out health services to non-governmental service providers to improve service utilization and health outcomes in rural, under-resourced communities [49]. One study covered by this review includes contracting out child health promotion services in Cambodia, which addressed two factors of disadvantages: residence (rural) and income (under-resourced communities). The review found low-quality evidence that the intervention had no effect on diarrhea incidence in children from these vulnerable groups. Therefore, it can be considered as having a neutral effect on health disadvantages.

*Financial and non-financial assistance and incentives (6 reviews).* Six reviews reported on the effects of financial and non-financial assistance and incentives to poor or vulnerable families and communities on child immunization coverage and uptake [62, 64, 72, 78, 83, 85]. One review by Owusu-Addo & Cross [83] also reported on the effect of conditional cash transfer on morbidity risks in children from selected infectious diseases. Two of these reviews were assessed as high methodological quality [62, 64], three as low [78, 83, 85] and one as moderate quality [72].

Among the factors of vulnerabilities addressed by these reviews, five covered interventions targeting poverty or low socio-economic status, with a particular focus on rural communities. Higher health needs were also found as factors of disadvantages in studies across all reviews.

Despite covering five of the same programs and four of the same primary studies, the findings of these reviews are mixed or even contradictory. The same trial from India—covered by three reviews [62, 72, 78]—shows moderate-quality evidence that food incentives can improve child immunization coverage. This would suggest that food incentives may be effective in addressing the health disadvantage these communities face in terms of immunization coverage. Referring to this trial, Jarrett et al. [72] pointed out that addressing the basic needs of the community as this intervention did, could be particularly relevant to building trust and expanding services in underserved and disadvantaged communities. Another study of non-monetary incentives at immunization visit covered by Johri et al. [62] also found this type of incentive effective in addressing the health disadvantage of the rural communities covered by improving immunization coverage.

Five reviews covered monetary incentives such as cash transfers or vouchers, with or without additional services [62, 64, 78, 83, 85]. Owusu-Addo & Cross [83] concluded that conditional cash transfers were effective both in improving child immunization coverage and in reducing morbidity risks from diarrhea and acute respiratory infections (ARI) in the targeted communities. Johri et al. [62] and Munk et al. [85] also concluded that selected types of incentives were effective in improving immunization coverage, although they highlighted a high risk of bias in the included studies and that there was heterogeneity in the results. In contrast, the reviews by Bright et al. [78] and Oyo-Ita et al. [64] found either mixed or no effect of monetary incentives on child immunization. The cause for these contradictions, including in the appraisal of the same evidence, is unclear. However, it should be noted that the reviews by Bright et al., Johri et al. and Oyo-Ita et al. [62, 64, 78] included a wider variety of programs, including non-conditional cash transfer and vouchers, which may partially explain the discrepancies and heterogeneity. The review by Oyo-Ita et al. [64] also noted that in three of the programs, children above two years old had no improvement in immunization coverage, thus suggesting a potential neutral or detrimental effect on inequalities between age groups. Looking specifically at the two high-quality reviews [62, 64], both highlighted the heterogeneity and uneven quality of the included studies and their findings. This may partially explain the contradictory conclusions these two reviews drew on the effect of incentive-based programs.

Therefore, we cannot conclude at this point on the effectiveness of such interventions in addressing health disadvantages.

**Promoting child health—Community, household and individual level interventions.** Of the forty-four reviews that cover interventions at the community, households and individual levels, twelve looked at interventions promoting child health. These covered two main types of interventions: nutrition supplements and health promotion education given to caregivers. Of the nine reviews covering nutrition supplements, six included interventions adopting a universal approach to health equity and four included interventions adopting a mixture of universal and targeted approaches. The reviews covering health promotion education to caregivers also adopted a mixture of universal and targeted approaches.

*Nutrition supplements (9 reviews).* Nine reviews addressed the effect of nutrition supplements on child morbidity and mortality from selected infectious diseases [48, 60, 61, 63, 66, 67, 70, 103, 107]. Seven of these reviews were assessed as high methodological quality [48, 60, 61, 63, 66, 67, 70] and two as critically low quality [103, 107].

Three of the reviews covered eighteen individual studies on zinc supplementation given in different formats and dosage to children, with or without other nutrition interventions [48, 63, 107]. Two reviews assessed the effect of this intervention on respiratory infection morbidity (including pneumonia) [63, 107], one on diarrhea incidence [107] and one on otitis morbidity [48] in the general population. Lassi et al. [63] and Tam et al. [107] reached opposite conclusions on the effect of zinc supplementation on respiratory infections in children. While Lassi et al. [63] found low-quality evidence that zinc supplementation reduced child morbidity from pneumonia, Tam et al. [107] found no significant effect of the intervention on lower respiratory tract infections (LRTI—including pneumonia, evidence quality unknown). Many factors may explain this contradiction, including the definition of the outcome observed, the quality of the reviews (high vs. critically low, respectively) or the year of the search and number of studies: Tam et al. is more recent and includes 10 new trials compared to Lassi et al. However, the lack of details on the studies included in Tam et al.'s meta-analysis makes further comparison difficult. Since Lassi et al. also highlighted the low-quality of the evidence available, we conclude that further research is needed to assess the effect of zinc supplementation on respiratory infections in children living in different contexts. Tam et al. [107] also assessed the effect of this intervention on diarrhea incidence and found a positive effect of zinc supplement on this outcome, although the meta-analysis showed high heterogeneity. Finally, Gulani and Sachdev [48] found no effect of the supplements on otitis morbidity in the general population. If we are to focus exclusively on the findings of high-quality reviews, there is low-quality evidence that zinc supplementation reduces pneumonia morbidity but has no effect on otitis [48, 63]. Assessing their impacts on health inequalities, these two reviews reported results for some vulnerable or disadvantaged populations. Lassi et al. [63] found low-quality evidence that zinc supplementation reduced child morbidity from pneumonia in specific groups such as populations living in slums, low-economic status neighborhoods or populations affected by HIV. Gulani and Sachdev [48] found studies which suggested a positive effect of the supplement for infants or malnourished children. However, Gulani and Sachdev [48] concluded that this evidence is currently insufficient to draw definitive conclusions on the benefits of zinc supplementation for specific vulnerable groups.

Three reviews [60, 61, 107] cover eleven individual randomized controlled trials (RCTs) of mixed quality giving vitamin A to neonates, infants or children alongside other vitamin supplements or selected vaccines. None of the reviews, including the two-high-quality reviews, found any effect from the interventions on either morbidity nor mortality from diarrheal infections, meningitis or respiratory infections (ARI or LRTI). The reviews did not report disaggregated effects of the interventions for different population groups, although the trials

captured by Haider et al. [60]'s review all took place in populations with high prevalence of either vitamin A deficiencies or HIV. Whether this review's finding could inform targeted nutrition interventions in these groups would require further research.

As for other micronutrient supplementation interventions in the general child population, Tam et al. [107] found no effect of multiple micronutrient (MMN) supplementation capsules or tablets, or of iron or iron/Folic acid supplementation on diarrhea morbidity. Finally, Eaton et al. [66] concluded that there was insufficient evidence to assess the effect of animal-source foods on infectious disease morbidity outcomes in infants and children across different settings. Neither reviews reported disaggregated effects of the intervention for different population groups.

Four reviews included nutrition supplementation interventions that adopted a mixture of universal and targeted approaches. Three reviews, covering two cross-sectional studies and twelve RCTs, examined the effects of point-of-use multiple micronutrient powder (POU-MNP) sachets or fortified flour provided to caregivers to be mixed with infants and children's food before consumption [67, 103, 107]. The findings of these reviews were mixed. While Carroll et al. [103] found that POU-MNP reduced diarrhea incidence amongst refugee children, Tam et al. [107] found evidence that the intervention actually increased diarrhea prevalence in healthy children. Meanwhile, Suchdev et al. [67] found no effect of POU-MNP in infants living in a variety of settings. The difference in settings and populations may explain these mixed findings. The reviews by Carroll et al. [103] and Suchdev et al. [67] included studies in vulnerable populations—such as children living in refugee camps, slums or malaria-endemic areas, while Tam et al. [107] focused on the general population. Looking at the only high-quality review, Suchdev et al. [67] concluded that the evidence was currently insufficient to conclude on the effect of POU-MNP on diarrhea morbidity or on other infectious diseases outcomes the review explored, such as malaria or upper respiratory tract infections. The fact that all three reviews acknowledge some contradictions or heterogeneity in their results further supports Suchdev et al.'s conclusions on insufficient evidence.

The last review looking at nutritional interventions is Imdad et al. [70]'s meta-analysis on neonatal oral probiotics/synbiotics supplementation added to breast milk and/or formula at various dosages and frequency. Although the review did not limit its search to specific groups of newborns, all twenty-one RCTs reporting on neonatal sepsis took place amongst low-birth-weight or preterm babies, primarily in hospital settings in middle-income countries. The authors found high-quality evidence that probiotics supplementation reduced the risk of sepsis amongst these vulnerable groups of neonates. Whether these findings could be extended to the general population or community settings requires further research. The review did not disaggregate the results further according to other factors of vulnerabilities or inequalities.

*Health promotion education to caregivers (3 reviews)*. Three reviews covering 13 individual studies addressed the effect of health promotion education to caregivers of children under five years old by health workers at community or individual levels [62, 78, 92]. Two reviews [62, 78], assessed as low and high methodological quality respectively, show mixed results on immunization uptake and coverage. Both reviews cover the same types of delivery mechanism (such community health workers, community groups, information session). Johri et al. [62], where both the review and the studies it covers were assessed as high quality, found these interventions had a positive effect and there was low heterogeneity among the studies covered. Conversely, Bright et al. [78], where both the review and its studies were assessed as lower quality, showed mixed results. It should be noted that several of these studies combined health promotion education with other interventions, which may explain the discrepancies. Finally, two studies covered by Flórez et al. [92] showed a positive effect of the intervention on morbidity from diarrhea and one study showed a positive effect on malaria morbidity. As all three

reviews brought together both targeted and universal interventions, it is difficult to draw definitive conclusions on the effect of these interventions on health inequalities.

## Protecting children from infectious diseases

We found two reviews covering a health protection intervention (Table 4). It was implemented at the household level and covered the impact of improved cookstoves to households [100, 105].

*Improved cookstoves (2 reviews).* Only two reviews, assessed as critically low quality, covered the effectiveness of improved cookstoves to reduce respiratory infection morbidity and mortality associated with household air pollution [100, 105]. Based on seven studies (six RCTs and a cohort study) covering almost exclusively rural communities in Latin America, Sub-Saharan Africa and one study in Nepal, this intervention had no effect on child morbidity from pneumonia or ARI of any severity. Due to the geographical and demographic specificities of the study population, the results may not be generalizable across different populations and countries. Whether the findings of these reviews may inform similar interventions with a targeted approach to population would have to be confirmed by further research. The reviews did not disaggregate the results further according to other factors of vulnerabilities or inequalities.

## Preventing infectious diseases in children

We found forty-eight reviews that looked at interventions aiming to prevent infectious diseases in children (Table 5). Seventeen covered interventions at the structural and policy intervention levels [49, 55, 58, 68, 81, 86–88, 90, 91, 93, 98, 101, 102, 104, 108, 110], three included multi-level interventions [62, 69, 97] and thirty-one looked at interventions at the community, households and individual levels [50, 54–57, 59, 64, 65, 69, 71–77, 79, 80, 82, 84, 85, 89, 90, 92, 94–97, 99, 106, 109].

**Preventing infectious diseases in children—Structural and policy levels.** Of the twenty-five reviews that covered interventions at the structural and policy levels, seventeen looked at infectious disease prevention interventions. These covered three main types of interventions: improving the water supply infrastructure, introducing new vaccines into national immunization programs and contracting out preventive health services.

The reviews covering water supply infrastructure and new vaccine introduction adopted a universal approach to health equity while the review on contracting out preventive health services adopted a targeted approach, focusing on a disadvantaged group in the countries of intervention.

*Water supply infrastructure (2 reviews).* Two reviews [55, 90], assessed as high and critically low methodological quality respectively, assessed the effect of improving water supply infrastructure on childhood diarrhea morbidity. Covering a total of six different studies, both reviews concluded that there was insufficient evidence to decide on the potential effectiveness of such interventions on childhood diarrhea risk. Neither of them reported results disaggregated by sub-groups or the effect of the interventions on inequalities.

*Introduction of new vaccines (14 reviews).* Fourteen reviews covered the effect of new vaccines, that have been introduced into national immunization programs [58, 68, 81, 86–88, 91, 93, 98, 101, 102, 104, 108, 110]. Among those, two covered Haemophilus influenzae type b (Hib) vaccines [86, 91], six covered pneumococcal conjugate vaccines (PCV) [58, 81, 87, 91, 104, 110] and eight rotavirus vaccines [68, 88, 91, 93, 98, 101, 102, 108]. With the exception of the review by de Oliveira et al. [58] and Soares-Weiser et al. [68], all these reviews were assessed as low [81] or critically low methodological quality [86–88, 91, 93, 98, 101, 102, 104, 110].

The Hib vaccine was found effective in reducing either Hib meningitis mortality [91] or morbidity [86] in infants (the group targeted by vaccination) in fourteen studies from Bangladesh, Pakistan and Brazil. DeAntonio et al. [91] also found that another study from Uruguay

suggested a potential herd effect of the introduction of the Hib vaccine among children under three years not targeted by the vaccine. However, the evidence was currently insufficient to confirm such an effect. The quality of the primary studies covered by the reviews was unclear and the reviews themselves were assessed as critically low quality, thus calling for caution in the interpretation of these results. The potential health inequalities effect of the Hib vaccine introduction is unknown, as neither review covering this vaccine reported any health inequalities measure.

The six reviews covering various types of PCV vaccines from thirty-one primary studies also found that further to their licensing and introduction into national programs, these vaccines were effective in reducing mortality [58] or morbidity [81, 87, 91, 104, 110] from pneumococcal infections in children. Bonner et al. [87], de Oliveira et al. [58] and Ngocho et al. [81] assessed the studies as fair to high quality. However, it should be noted that half of these reviews focus exclusively on Latin America [58, 87, 91] while the other half focused on the African continent only [81, 104, 110]. Hence the generalizability of these results across all LMICs and continents cannot be confirmed. Looking at the health equity implications of PCV vaccines, five of the reviews included some measure of health inequalities. Three reviews [58, 81, 104] found a stronger decline of pneumococcal diseases in younger children, the group carrying a higher burden of the disease, suggesting that the vaccine may help reduce inequalities between age groups in the burden of these diseases. The review by Bonner et al. [87] also focuses on children above the vaccination age and concludes that the implementation of catch-up doses of PCV vaccines for children above the targeted age at the time of vaccine introduction was effective in reducing the burden of pneumococcal disease. However, the review by Vardanjani et al. [110] highlights that PCV vaccines may have a detrimental impact on health inequalities in another domain. Reviewing case-control studies from South Africa bringing together HIV infected and uninfected children, Vardanjani et al. not only found that vaccine effectiveness was lower amongst HIV positive children but that this effect may also reverse the positive effect on the gap between age groups. Whether these conclusions can be extended to settings and HIV-affected populations outside of South Africa would require further research. Looking at the only high-quality review covering PCV vaccine, findings by de Oliveira et al. [58] confirm a positive effect of the PCV vaccine on pneumococcal meningitis mortality and on the gap between younger and older children.

Finally, all eight reviews covering different types of rotavirus vaccines found the vaccine effective in reducing child mortality [88, 91, 93, 98] or morbidity [68, 88, 93, 98, 101, 102, 108] due to diarrhea or gastroenteritis, with or without confirmation of rotavirus infection. These eight reviews covered 104 individual primary studies. DeAntonio et al. [91] also found evidence suggesting a potential herd effect of the vaccine among children under five years above the vaccination age but concluded that there was currently insufficient evidence to confirm this finding. However, all but one of these reviews were assessed as critically low quality and the quality of the primary studies they covered was also mixed. Four out of six of these reviews were also exclusively focused in Latin America [88, 91, 98, 101], hence the number of overlaps in the vaccination programs and studies covered. This region is therefore over-represented in the results. Yet, Soares-Weiser et al. [68], the only high-quality review for this intervention, found moderate- to high-quality evidence that all three types of rotavirus vaccines approved by WHO were effective in reducing the burden of rotavirus or all-cause diarrheal diseases across fifteen low- and middle-income countries. However, the meta-analyses showed statistical moderate to high heterogeneity and variations in the impact of the vaccine according to the type of diarrheal disease considered. Looking at the health equity implications of these interventions, four reviews noted differences in the effectiveness of the vaccine across age groups and/or contexts and one review reported disaggregated results for selected vulnerable groups.

Three reviews noted a higher decline of gastroenteritis morbidity/mortality among infants, the age group carrying a higher disease burden, suggesting a potential reduction of disease burden inequalities between age groups [68, 88, 102]. Three reviews also highlighted both uneven efficacy and effectiveness of the vaccination between countries and regions [88, 93, 102], with Chavers et al.'s [88] results suggesting that the vaccine was most effective in countries with comparatively lower child mortality. These findings may point towards the potential increase in global inequalities. Yet, the mixed quality of evidence available and uneven geographical coverage of the reviews suggest that there is currently insufficient evidence to assess the effect of rotavirus vaccines on global health inequalities. As for the effect of rotavirus vaccination on disadvantaged groups, the review Soares-Weiser et al. [68] reveals opposite results depending on the factor of disadvantage considered. While rotavirus vaccination significantly reduced the burden of diarrheal diseases amongst malnourished children, it had no significant effect amongst HIV-exposed or -infected children. However, the number of studies supporting these findings was very small, with one RCT reporting on malnourished children and three on HIV-exposed or -infected children.

*Contracting out preventive services (1 review).* The review by Odendaal et al. [49] on contracting out health services to non-governmental service providers also covered the contracting out of preventive services. The authors found moderate-quality evidence that such an intervention had no effect on immunization uptake in children in neither country of intervention—Cambodia and Guatemala. This suggests that this intervention had a neutral effect on health disadvantages when it comes to immunization uptake.

**Preventing infectious diseases in children—Community, households and individual levels.** Of the forty-four reviews that cover interventions at the community, households and individual levels, thirty-one included infectious disease prevention interventions. The reviews looked at interventions aiming to increase the use or coverage of preventive tools (immunization, bed nets, infant HIV prophylaxis) as well as water, sanitation and hygiene (WASH) interventions, and hospital-based interventions to prevent infections in vulnerable neonates.

The reviews covering WASH interventions adopted a universal approach to health equity while the reviews on hospital-based interventions towards vulnerable neonates adopted a targeted approach. Finally, the reviews covering interventions aiming to increase the use or coverage of preventive tools synthesized together interventions with targeted and universal approaches to health equity.

*Increasing child immunization (17 reviews).* Seventeen reviews covered interventions at community or individual levels aiming to improve child immunization uptake and/or coverage either by increasing the demand for or the supply of immunization services [50, 59, 64, 65, 72–74, 82, 85, 89, 92, 95–97, 99, 106, 109]. These reviews synthesized 100 individual primary studies, almost all of which were from Asia and Sub-Saharan Africa, with only three studies from Latin America. Three of these reviews were assessed as high quality [59, 64, 65] and three as moderate [72–74]. The other eleven were assessed either as low [50, 82, 85] or critically low methodological quality [89, 92, 95–97, 99, 106, 109].

All fourteen reviews covering intervention aiming to increase the *demand* for child immunization -such as immunization education, communication, social mobilization and information campaigns, and reminder tools with or without outreach services—found some positive effect on the coverage or uptake of one or more vaccines [50, 64, 65, 72–74, 82, 85, 89, 92, 95–97, 106]. However, eight of them also reported at least one study or outcome where the intervention was ineffective [65, 74, 85, 89, 95–97, 106] and seven meta-analyses reported moderate to high heterogeneity in their results [64, 65, 73, 74, 95, 96, 106]. Neither the type of vaccine outcome measured nor the sub-type of intervention seem to explain alone these variations. The quality of the primary studies was also very heterogeneous. A sub-analysis of high- and

moderate-quality reviews leads to the same findings, namely, that while this type of intervention is found effective in improving childhood vaccine uptakes and coverage in most studies, all five reviews raised the issues of the quality of the evidence base and the heterogeneity of pooled results [64, 65, 72–74]. In terms of health equity, three of these reviews discussed the effect or suitability of these interventions among different groups or settings [50, 72, 97] and six reviews focused specifically on a disadvantaged group [64, 65, 72, 82, 89, 97]. The factors of disadvantage targeted by these interventions included higher health needs (especially low vaccination rates) [64, 65, 72, 82], poverty [64, 65, 72, 89], place of residence and type of dwelling (e.g., rural communities, urban slums, informal settlements or small dwellings) [50, 64, 65, 89, 97], ethnicity [97], and low literacy or education [50, 64]. Overall, the findings suggest that such interventions may be effective both at population level and in addressing the targeted populations' health disadvantage. However, the results' heterogeneity warrants further research to confirm such claim. The type of interventions covered also seems to affect their impact on health equity. For example, three reviews highlighted targeted communication or education towards pre-identified disadvantaged groups as an effective approach to raising immunization uptake in those groups [50, 72, 97]. Conversely, the evidence on the potential effect of mass communication promoting immunization on health inequalities is less clear. While Yuan et al. [50] concluded that the evidence was unclear and limited (only one study), Jarrett et al. [72] highlighted the fact that such 'top-down' approaches to communication may not be effective across all groups, depending on the level of vaccine hesitancy–thus potentially raising inequalities issues.

The eight reviews reporting on interventions targeting the *supply* of child immunization services, found mixed results, with variations in effectiveness according to the specifics of the intervention, the type of vaccine considered or even amongst a similar type of intervention [59, 64, 72, 82, 85, 97, 99, 109]. The quality of the primary studies included was also uneven [59, 64, 72, 82, 85, 97, 99]. They covered interventions such as the integration of immunization with other health services, improving community or outreach services, the introduction of new technology or strategies, the provision of tools and material to support immunization health professionals or health professional training, either on their own or as part of multi-component interventions. The findings of the three high- and moderate-quality reviews were consistent with those highlighted above. In terms of health equity, two of the eight reviews reported on the comparative effect of service integration interventions in rural versus urban areas [97, 99]. Smith et al. [99] reported on one poor-quality study showing that while the integration of HIV and immunization services had a positive effect on overall population immunization uptake in urban sites, its effect was detrimental in rural sites due to stigma and concerns over discrimination. Nelson et al. [97] also highlighted the potential risk of increased rural/ urban disparities with integrated services, reporting on one low-risk-of-bias study showing that such integration could not address distance issues, thus making it more suitable and effective to urban contexts rather than rural ones. Both studies took place in Sub-Saharan African countries. While the evidence from these reviews is mixed, they do point towards a potential risk for this type of intervention to increase inequalities in service uptake between rural and urban areas. As for the impact of supply-side interventions on addressing the health disadvantage, the five reviews reporting on interventions adopting a targeted approach to health equity suggested that the effect of these interventions on addressing various health disadvantages would require further high-quality studies to be confirmed [64, 82, 97, 99, 109]. Four reviews focused on areas or communities with higher health needs (e.g., low vaccination rate or high infectious disease prevalence). The other factors of disadvantages addressed by these interventions were the place of residence (i.e., urban versus rural communities) [64, 97, 99, 109], poverty [64] or families identified as vulnerable in their individual context [99, 109].

*Increasing bed net use (3 reviews).* Three reviews covered interventions aiming to increase bed net use for children under five [54, 76, 96]. The review by Naugle & Hornik, assessed as critically low methodologically, reported on a single study from Cameroun covering the effect of a mass media campaign to promote consistent bed net use [96]. It concluded that the evaluation of this campaign provided strong evidence that this intervention was effective in increasing bed net use for children under five. This review only reported the overall population impact of the intervention and did not report any disaggregated effect of the invention on different groups.

Two reviews, assessed as high and moderate quality methodologically, synthesized six studies on community outreach education activities on the use of insecticide treated nets (ITNs) and community health services to mothers and households with children under five [54]. All but one study took place in Sub-Saharan African countries, in malaria-endemic areas. The reviews found low-quality evidence that such interventions may be effective in increasing under-5 use of ITNs in these communities. This suggests that this intervention may reduce the health disadvantage faced by communities in malaria-endemic areas regarding the burden of that disease.

*Preventing HIV transmission (3 reviews).* Three reviews assess the effects of preventive HIV services for a targeted group of infants—those exposed to HIV, with the HIV status of the mother as the main factor of disadvantage [75, 84, 99]. One review focuses on HIV prophylactic treatment given to breastfed, HIV-exposed infants [75] while the other two focus on *how* HIV prophylactic treatment is delivered to HIV-exposed infants. All but one of the twelve included studies occurred in Sub-Saharan African countries.

The moderate-quality review by White et al. [75] found that compared to standard care, extending HIV infant prophylaxis throughout the breastfeeding period reduces the rate of HIV infections amongst HIV-exposed infants, even when the mother is not receiving antiretrovirals. While these findings are based on a small number of studies in urban sites, the authors assessed the evidence as moderate to high quality, thus recommending that HIV infant prophylaxis must continue until weaning.

The two reviews by Puchalski Ritchie et al. [84] and Smith et al. [99], assessed as low and critically low methodologically respectively, show mixed results on the uptake and coverage of preventive HIV services for infants exposed to HIV [84, 99]. Both reviews suggested that home visits by community health workers may be effective in addressing the health disadvantage faced by infants born to HIV positive mothers, leading to an increase in uptake of HIV prophylaxis. However, the quality of the two primary studies reporting on this type of intervention was mixed.

As for the integration of prevention of mother-to-child transmission (PMCT) with mother and child healthcare services, the reviews drew conflicting conclusions. Puchalski Ritchie et al. [84] found this intervention had either no or even detrimental effect on HIV prophylaxis uptake at birth from two trials assessed as having a high risk of bias. Smith et al. [99] found fair to good-quality evidence that integrating PMCT into post-partum care or immunization services increased the uptake of HIV prophylaxis. It is unclear whether the setting or the study design may explain these differences. This leads us to conclude that the effectiveness of service integration in addressing some of the health risks of HIV exposed children is currently unclear.

Puchalski Ritchie et al. [84] also covered three other interventions targeting HIV positive mothers and their infants, including peer-to-peer education, task-shifting and service quality improvement, none of which showed any effect on HIV prophylaxis uptake at birth. This suggests that these interventions seem to have no effect on addressing the health disadvantage of HIV exposed infants. The authors also noted the high risk of biases of the three trials testing these interventions.

*WASH interventions (7 reviews).* Seven reviews covered WASH interventions at the community or household level from 90 individual studies, reporting on their impacts on the overall population [55, 69, 77, 79, 80, 90, 94]. Two reviews were assessed as high quality [55, 69], one as moderate review [77] and four reviews as low [79, 80] or critically low methodologically [90, 94].

Three reviews covered 39 individual studies on water quality improvement at point-of-use [55, 77, 90]. Despite the difference in the quality of each review (rating from critically low to high), all three reviews found very similar results for individual type of interventions (i.e., chlorination, flocculation, solar disinfection). They concluded that there was low- to moderate-quality evidence that these interventions were effective in reducing diarrhea morbidity in children under five years old at population level. However, the reviews also highlighted the high heterogeneity of the results. None of the reviews reported disaggregated results by population groups for these types of interventions, thus providing no information on the potential impact of the intervention on health inequalities. However, Soboksa et al. [77]'s analysis by world regions shows significant differences in the effect of solar disinfection between Latin American and Asian countries, thus raising questions as to the potential implications of such interventions for global health inequalities.

Five reviews covered sanitation and hygiene interventions, such as hygiene promotion and education, community mobilization and campaigns, with or without the development of a WASH infrastructure [69, 79, 80, 90, 94]. These reviews showed mixed results of these interventions on morbidity from selected infectious diseases in children under five. Of the four reviews reporting on the effect of sanitation and/or hygiene interventions on diarrhea morbidity, only one review found them effective and reported high-quality evidence supporting their conclusions [79]. The other three showed mixed results depending on the context and the details of the intervention, also highlighting the low quality and lack of conclusive evidence [69, 90, 94]. The findings of the only high-quality reviews covering these types of interventions were consistent with those highlighted here [69].

The two reviews reporting on respiratory infections and pneumonia also had mixed findings. One review by Morita et al. [94] found no conclusive evidence on the effect of hygiene interventions on this outcome while the second by McGuinness et al. [80] showed mixed results. Both reviews highlighted the low quality of evidence currently available and were assessed as low and critically low quality themselves. However, McGuinness et al. [80] highlighted the result of one high-quality cluster RCT from Pakistan that had found hygiene education and the provision of hygiene product effective in reducing pneumonia illness rate. Finally, three reviews explored the effect of these interventions on other infectious diseases [69, 79, 94]. With the exception of trachoma cases–for which Freeman et al. [79] found moderate to high quality evidence that sanitation and hygiene interventions were effective in reducing trachoma morbidity, the reviews found either no effect [69, 79] or insufficient evidence [94] that sanitation and hygiene interventions had any impact on parasitic diseases; or on dysentery prevalence [69].

While some of this heterogeneity in results may be explained by variations in intervention settings, targeted population and the quality of evidence available, these variations also offer some findings relevant to health inequalities research. McGuinness et al. [80] highlighted that the effectiveness of hygiene interventions varied according to intervention settings and characteristics (urban child care-based vs. rural home-based interventions) as well as compliance but was unable to attribute these variations to a specific factor with available evidence. Majorin et al. [69] reported on one evaluation where the education and hygiene promotion intervention showed a reduction of diarrhea prevalence in rural sites but not in urban slums. Morita et al. [94] suggested that the effect of these interventions vary according to the child's age. As

only Majorin et al. [69]'s review was assessed as high quality, a sub-analysis according to the quality of the reviews cannot provide further insights into the implications of these interventions for health inequalities. Hence, while some of these reviews suggest that hygiene and sanitation interventions may be relevant when studying health inequalities in the burden of child infectious diseases, the evidence is currently insufficient to conclude on their effect on health inequalities or the relevant factor of inequalities they affect.

*Preventing infections in vulnerable neonates (2 reviews).* Two reviews, for which were assessed as high quality, synthesized the effect of targeted interventions in hospital settings to reduce the risk of infections in low-birth weight or premature neonates [56, 57]. The higher health needs of low-birth weight or premature babies was the main factor of disadvantage covered in these reviews. As all the interventions took place in hospital settings, including neonatal specialized units, the generalizability of the results beyond this specific setting is not possible.

Conde-Agudelo & Diaz-Rossello [57] included seven RCTs from India, Colombia, Ecuador and Madagascar relevant to this umbrella review. The review assessed the effect of kangaroo mother care on the risk of illnesses (as defined in individual studies) or selected infectious diseases at six months follow-up as compared to routine care. Based on these trials, the authors found moderate-quality evidence that this intervention had a positive effect the risks of severe illness, nosocomial infection/sepsis and lower respiratory tract diseases but no effect on mild/moderate infection and illness or diarrhea. Thus, this intervention may be effective in addressing the higher health risks of selected infectious diseases faced by this vulnerable group of infants.

The review by Cleminson & McGuire [56] assessed the effect of topical application of moisturizing emollients (ointments, creams, or oils) to increase the protective barrier function of the skin in premature infants and so, prevent infections in this vulnerable group. Of the synthesis of six trials relevant to this review, the authors found no effect of such an intervention on the incidence of invasive infections, regardless of the type of emollient used. Therefore, this intervention does not seem effective in addressing the higher health needs of this vulnerable population.

**Preventing infectious diseases in children—Multilevel interventions.** We found four reviews [62, 69, 85, 97] covering preventive, multilevel interventions. All of them reported the overall population impacts of these interventions–with no equity aspects.

*Combined demand-and supply-side interventions for child immunization (3 reviews).* Three reviews [62, 85, 97] covering ten individual studies assessed the effect of combined demand- and supply-side interventions across levels to improve child immunization coverage and/or uptake. The quality of the reviews themselves were uneven as Johri et al. [62] was assessed as high methodological quality while Munk et al. [85] and Nelson et al. [97] were appraised as low and critically low respectively. Despite these differences, all three reviews found these interventions effective in improving child immunization but highlighted the heterogeneity of the result and issues with the quality of some of the studies.

While these interventions themselves did not all follow a targeted population approach by design, the reviews were focusing on urban populations [97], rural areas [85] or communities with lagging health or social indicators [62]. Hence the findings may only be relevant to these groups specifically. Then, these findings may inform targeted approaches addressing the health disadvantage of urban populations or communities with lagging health or social indicators regarding immunization. However, further research would be needed to confirm it. None of the reviews disaggregated their results by subgroup among the study population.

*Combined hygiene education and water supply/sanitation infrastructure provision (1 review).* The high-quality review by Majorin et al. [69] synthesized the results of two studies in

Bangladesh where rural populations received an intervention combining water and sanitation infrastructure development with hygiene education and water quality improvement. The review found very low-quality evidence that such a combined intervention reduced dysentery and diarrhea prevalence.

## Discussion

This umbrella review explored which public health interventions are effective in reducing morbidity, mortality and health inequalities from infectious diseases amongst children in LMICs. We found sixty systematic reviews synthesizing 453 individual primary studies. Yet, only twenty reviews provided results on intervention effectiveness across different groups while another twenty-one reported their effectiveness on the targeted disadvantaged groups. Universal approaches to health equity were found in interventions covered by forty-five (75%) of the reviews while twenty-six reviews (over 40%) included interventions adopting targeted approaches. No review covered interventions following a proportionate universalism approach. These trends are in line with those from previous global umbrella reviews [14, 16], thus highlighting once again the need for a stronger equity lens and systematic reporting of intervention effects across population groups in systematic reviews.

Reviews covering preventive interventions at community, household or individual levels were overwhelmingly represented in our review. Hence, proportionally less evidence is available regarding interventions at structural or policy levels or health promotion and protection interventions. Evidence on effective health protection interventions is particularly weak as no such interventions was identified at any policy levels. While how we defined our outcomes of interest may have contributed to this unbalance, this may also result from the difficulty to conclusively link high level or broad health interventions to changes in the burden of specific diseases.

### Main findings on public health interventions reducing morbidity, mortality and health inequalities from infectious diseases amongst children in LMICs

Based on their effect, interventions can be grouped in five categories. The first one includes interventions that may be effective both at population level and in addressing health disadvantages or gaps. This includes communication, education, social mobilization and outreach interventions to increase the use of certain preventive tools such as immunization or bed nets. However, the heterogeneity of some of the results suggests that their effectiveness also depends on the intervention approach and population targeting adopted, and the specific context of implementation.

A second group covers interventions that are or seem effective at population level but requires further research, especially regarding their effect(s) on health inequalities. These include health promoting interventions (such as health promotion education to caregivers and oral probiotics/synbiotics supplementation), multilevel interventions to promote child immunization or WASH interventions and selected preventive interventions (such as the introduction of new vaccines into national programs and water quality improvement interventions at community level). These interventions would benefit from implementing a stronger equity lens during their evaluation and a more systematic reporting of their impact across different population groups.

In the third group, we find interventions for which the evidence base on their effectiveness is mixed or even contradictory both at population level and on specific groups. These include financial assistance and incentives (such as monetary incentives), selected nutrition

interventions (such as point-of-use multiple micronutrient powders), sanitation and hygiene interventions at community or household level, and interventions targeting the supply of child immunization services (such as service improvement, service integration and healthcare professional training). These interventions would require further, high-quality research to better understand their effects in different settings and the reasons behind this heterogeneity.

A fourth group includes interventions found effective in addressing the health disadvantage of infants at higher risks of infection but which the potential effect on the wider burden of infectious diseases in their community is unknown. These include community health workers home visits to HIV positive mothers after birth and kangaroo mother care for low-birth weight and premature babies in hospital settings.

Finally, the fifth group includes interventions that have been found ineffective on any of the outcomes of interest in this review. These include contracting out health services to non-governmental service providers for under-served communities, several nutrition supplementation interventions (such as Vitamin A, iron, multi-micronutrient tablets and capsules, or animal-source foods), improved cookstoves to reduce indoor air pollution, emollients to prevent infections in premature infants as well as HIV peer-to-peer education or HIV service improvement and task shifting.

Three types of preventive intervention–selected new vaccines introduced into national immunization programs, sanitation and hygiene interventions, and HIV services integration with other health services–bring to our attention on how an intervention may have a positive effect according to one factor of health inequalities (e.g., age or health risk) but may be detrimental according to another (e.g., place or country of residence). This highlights the importance for health inequalities research to report results disaggregated across several socio-demographic dimensions. Research on the introduction of rotavirus vaccines and on water quality improvement at point-of-use also points towards a risk of increasing global health inequalities, which raises the question of the multilevel nature of health inequalities.

## Implication for research, policy and practice

This umbrella review contributes to fighting infectious diseases and improving child health in several ways. First, this review can inform strategic decisions about research and research funding. By identify the areas and outcomes where research is most needed, it can help prioritize research in public health and equity in LMICs.

Secondly, this review can help guide the choice of practitioners and policy makers towards interventions with proven effectiveness. Indeed, this review identifies interventions where evidence is the strongest to address either a specific condition (e.g., diarrheal morbidity) or a series of a health challenges with a single intervention (e.g., WASH or nutrition interventions). At the same time, it can help identify areas or interventions where, due to the current state of the evidence, an intervention may require further adaptations, pilots or experimentation before being implemented in a new context or scaled up. For example, the evidence behind interventions targeting HIV-exposed infants or the adoption of the Hib or PCV vaccines described in this review has been provided by studies from only a handful of countries. Meanwhile, the evidence supporting selected WASH multilevel interventions or nutrition supplementation–although promising–is still insufficient.

Finally, this review can help inform discussions in a context of competing child health priorities. By exploring the effect of individual interventions both at the population level and on health inequalities, we have further demonstrated the complex relationship between population health and health equity goals. For example, we found that interventions addressing the demand for immunization at the lower policy level can offer a double gain at population level

and amongst several disadvantaged groups. At the same time, selected interventions such as child health preventative services integration might be beneficial to one group but not another.

It should be noted again that this review was focused on specific outcomes affecting morbidity, mortality and health inequalities from infectious diseases amongst children in LMICs. Therefore, the findings presented here should be assessed and understood within the broader body of public health evidence addressing the needs of children but also other populations or affecting other causes of ill-health. Additionally, the findings of this review should be interpreted within the specific burden of infectious diseases faced by an area, country or region. To provide this broad overview of the field and identify interventions that can offer health gains on multiple diseases outcomes, we purposely refrained from a disease- or transmission mode-specific approach. Yet, these are essential components in the choice of intervention, its design and its chances of success. Hence, the findings of this review should be carefully interpreted according to the specific context and public health system in which they are to be used.

## Strengths and limitations of the evidence base

The low methodological quality of the reviews available remains a major barrier to the establishment of a strong evidence base in this field. Although we adopted a strict definition of what would qualify as a systematic review, more than half of the included reviews were assessed as low- or critically low-quality. While some of the weaknesses commonly identified in these reviews may be due to resource constraints (e.g., extraction in duplicate), the two most common ones affect information reporting on excluded or included studies, thus highlighting the need for further implementation of existing good practices and checklists. For complex interventions or reviews covering a wide range of interventions, such weaknesses also limit our ability to effectively compare the findings of different reviews or explore potential reasons for contradictory findings. However, for certain types of interventions–such as interventions to increase immunization at the community, household or individual levels–the reviews themselves were limited by the low quality and heterogeneity of the primary studies they covered. While this limitation *does* call for further strengthening of the methodology and the application of existing guidelines for public health intervention research, it may also reflect the challenges inherent to assessing real world interventions and natural experiments rather than RCTs alone. An example of these challenges was the issue raised by certain reviews–such as those covering sanitation interventions or multi-level interventions for immunization–on the difficulty of assessing the effectiveness of specific components that were part of wider, complex interventions.

A second challenge comes from the public health field itself. Heterogeneity of interventions, study designs and results according to context was a recurring challenge raised by the reviews we included. This is further illustrated by the number of reviews including narrative synthesis alongside, or instead of, meta-analysis. This challenge is common in umbrella reviews covering public health interventions [111]. It has also driven our decision not to attempt a meta-analysis or compare the effectiveness of various interventions. While this difficulty does not affect the validity of our conclusions, it calls for careful consideration for the context and intervention design when applying evidence in a specific setting.

## Strengths and limitations of this umbrella review

The scope of our umbrella review was broad in order to provide an exhaustive overview of available evidence on interventions able to address the burden of infectious diseases and related health inequalities amongst children in LMICs. Our umbrella review was based on a comprehensive search of both academic and grey literature on a wide range of interventions.

To address the difficulties raised by previous research in identifying systematic reviews with an equity lens, we adopted a broad definition of health inequalities as variations between groups and did not include any health inequality-related terms in our search strings. While our searches were carried out in English, we did not exclude any languages in our screening thus making sure to capture research from all regions of the global South. Finally, while our definition of systematic reviews and evidence synthesis according to the DARE criteria may have led to the exclusion of certain records covering relevant interventions, it has ensured that the present review captured the best available evidence on how to reduce child mortality, morbidity and health inequalities due to infectious diseases in LMICs.

## Conclusion

Building a strong evidence base on public health intervention adapted to LMIC context is essential to inform policy and reduce health inequalities while improving child health at population level. This umbrella review aimed to respond to that need by synthesizing the best available evidence on public health interventions effective in reducing mortality, morbidity and health inequalities from infectious diseases amongst children under five years old living in LMICs. While this review identified selected interventions providing solid evidence to respond to this challenge, we also identified a number of gaps, especially regarding their implications for health equity.

We found that communication, education, social mobilization and outreach interventions are effective in improving the use of preventive tools like immunization or bed nets both at population level and in addressing the health needs of the most disadvantaged. Such approaches offer a strong avenue to reduce morbidity and mortality from infectious diseases in children under five years old.

In contrast, we identified a number of health promoting, health protecting and preventive interventions that are not effective in reducing child morbidity and mortality from infectious diseases or addressing the health of disadvantaged populations. These include contracting out health services to non-governmental service providers for under-served communities, selected nutrition supplementation to infants and children, improved cookstoves to reduce indoor air pollution, emollients to prevent infections in premature infants as well as HIV peer-to-peer education or HIV service improvement and task shifting.

Finally, while none of the interventions covered seem to be detrimental to child health at the overall population level, some raise concerns as to their potential health equity implications. These include health service integration, sanitation and hygiene intervention and rotavirus and PCV vaccine introduction. Further research is required to confirm these findings across various factors of health inequalities. These interventions also highlight the importance for health inequalities research to report results disaggregated across several socio-demographic dimensions and consider the equity implication of an intervention not only locally but across multiple levels.

Our review confirms the need for further, high-quality research in LMICs on the effects of public health interventions at both the overall population level and especially in terms of reducing health inequalities. We also found a large gap in the evidence base on the effectiveness of health protection interventions, which aim at safeguarding children from the risk of infectious diseases through legal, regulatory or enforcement mechanisms.

## Supporting information

**S1 Appendix. Intervention table.**
(DOCX)

**S2 Appendix. Published protocol.**
(PDF)

**S3 Appendix. PRISMA-E.**
(PDF)

**S4 Appendix. Pilot literature search.**
(DOCX)

**S5 Appendix. Full search strategy.**
(DOCX)

**S6 Appendix. Extraction template.**
(DOCX)

**S7 Appendix. List of excluded studies.**
(DOCX)

**S8 Appendix. Primary study citation matrix.**
(XLSX)

**S9 Appendix. Description of included reviews.**
(DOCX)

## Acknowledgments

The authors would like to thank Professor Terje Andreas Eikemo for his comments and feedback on the draft version of this manuscript, as well as Mrs. Nataliia Korotkova, Miss Cindy Duvivier and Doctor Omid Rasouli for their help with article translation. The authors would also like to thank Marte Ødegaard, from the Library of Medicine and Science at the University of Oslo for her assistance with selected database searches.

## Author Contributions

**Conceptualization:** Elodie Besnier, Katie Thomson, Adam Todd, Clare Bambra.

**Formal analysis:** Elodie Besnier.

**Investigation:** Elodie Besnier, Katie Thomson, Donata Stonkute, Talal Mohammad.

**Methodology:** Elodie Besnier, Katie Thomson, Nasima Akhter, Adam Todd, Magnus Rom Jensen, Astrid Kilvik.

**Project administration:** Elodie Besnier.

**Supervision:** Katie Thomson, Nasima Akhter, Adam Todd, Clare Bambra.

**Validation:** Katie Thomson, Donata Stonkute, Talal Mohammad.

**Visualization:** Elodie Besnier.

**Writing – original draft:** Elodie Besnier.

**Writing – review & editing:** Elodie Besnier, Katie Thomson, Donata Stonkute, Talal Mohammad, Nasima Akhter, Adam Todd, Magnus Rom Jensen, Astrid Kilvik, Clare Bambra.

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
