## [Decision Letter · Decision Letter 0]

23 Dec 2020

PONE-D-20-17157

Which public health interventions are effective in reducing morbidity, mortality and health inequalities from infectious diseases amongst children in low- and middle-income countries (LMICs): an umbrella review.

PLOS ONE

Dear Dr. Besnier,

Thank you for submitting your manuscript to PLOS ONE. After careful consideration, we feel that it has merit but does not fully meet PLOS ONE’s publication criteria as it currently stands. Therefore, we invite you to submit a revised version of the manuscript that addresses the points raised during the review process.

Please address the requested changes to the best of your knowledge, incluiding a justification of augmenting the review framework. Because low and critically low quality studies were included in several analyses, a separation of high and low quality studies findings would make the manuscript easier to understand. 

We look forward to receiving your revised manuscript.

Kind regards,

Abraham Salinas-Miranda, MD, PhD

Academic Editor

PLOS ONE

Journal Requirements:

2. Please clarify whether bias and heterogeneity of studies were evaluated in this review.

3. At this time, we ask that you please update the search period in your search strategy in order to include studies that have been published within the last 12 months. Please update your manuscript accordingly to include these newly published studies. Thank you for your attention to this request.

4. We note that Figure 4 in your submission contain map images which may be copyrighted. All PLOS content is published under the Creative Commons Attribution License (CC BY 4.0), which means that the manuscript, images, and Supporting Information files will be freely available online, and any third party is permitted to access, download, copy, distribute, and use these materials in any way, even commercially, with proper attribution. For these reasons, we cannot publish previously copyrighted maps or satellite images created using proprietary data, such as Google software (Google Maps, Street View, and Earth). For more information, see our copyright guidelines: http://journals.plos.org/plosone/s/licenses-and-copyright.

4.1. You may seek permission from the original copyright holder of Figure 4 to publish the content specifically under the CC BY 4.0 license. 

4.2. If you are unable to obtain permission from the original copyright holder to publish these figures under the CC BY 4.0 license or if the copyright holder’s requirements are incompatible with the CC BY 4.0 license, please either i) remove the figure or ii) supply a replacement figure that complies with the CC BY 4.0 license. Please check copyright information on all replacement figures and update the figure caption with source information. If applicable, please specify in the figure caption text when a figure is similar but not identical to the original image and is therefore for illustrative purposes only.

Reviewers' comments:

Reviewer's Responses to Questions

**Comments to the Author**

1. Is the manuscript technically sound, and do the data support the conclusions?

Reviewer #1: Yes

Reviewer #2: Yes

2. Has the statistical analysis been performed appropriately and rigorously? 

Reviewer #1: Yes

Reviewer #2: Yes

3. Have the authors made all data underlying the findings in their manuscript fully available?

Reviewer #1: Yes

Reviewer #2: Yes

4. Is the manuscript presented in an intelligible fashion and written in standard English?

Reviewer #1: Yes

Reviewer #2: Yes

5. Review Comments to the Author

Reviewer #1: The authors sought to investigate the effectiveness of public health interventions in reducing morbidity, mortality and health inequalities in children younger than 5 years with infectious diseases by using an umbrella review.

This is a well written manuscript. The narrative of the methods flows smooth and the reader is confident of the results because of the detailed description of the methods.

Page 21 line 258 Quality of the evidence: because low and critically low quality studies were included in several analyses, the manuscript will be enhanced if at the end of each section the authors will include only high and moderate quality studies (resembling a sensitivity analysis). I see a summary in some of the sessions at the end of the manuscript but not at the beginning.

Page 33 line 420 Protecting children from infectious diseases. The authors did not find any review that protected specifically children from infectious diseases and approached population and health inequalities. I am wondering if the reviewers included all interventions guided to promote vaccination. Clarifying this point will be helpful.

Reviewer #2: Abstract

The abstract clearly summarizes the review and provides a succinct narrative of the study’s conclusions. Minor suggestions are:

1. The authors stated in the study objectives that the focus of the umbrella review was for the purpose of “identify public health interventions that are effective in reducing mortality, morbidity and health inequalities from infectious diseases in low- and middle-income countries. This seems like a reach in terms of the findings and discussions of the paper.

2. Line 28, should be more specific in line with the aim of the manuscript

3. Line 29 says to address these gaps, but only one gap was mentioned in the previous statement(s).

Introduction

Overall, the authors do a good job of providing background information that supports the need of the study. The literature review adequately addresses existing literature on the topic (including relevant definitions of numerous terms). Recommendations for the introduction are as follows:

1. Lines 55 – 60. There is established evidence that education is related to income, but this relationship seems to be lost as the authors skip from talking about income to talking about education. A bridging sentence to explain the correlation between income and education will explain the cross from income to education.

2. Living in poor households, offspring of mothers with no education and inequalities for children living in rural areas are considered deprived groups, therefore I suggest that the “Inequalities are also found within deprived groups” found on line 60 be moved to line 56

3. The statement on line 65 requires a citation.

4. Lines 66 – 70 talks about leading causes of death for children in LMIC, I suggest that rather than have multiple sentences talking about the same issue, group the Pneumonia, diarrheal disease, Malaria, and HIV together in one sentence. The addition of the DALYs in those lines does not seem to add further information to the manuscript and I suggest it should be deleted.

5. Line 71, what diseases did the authors consider? The authors should be specific what types of infectious diseases they looked at.

Methods

The authors do a thorough job of presenting their rationale in describing their framework, excluding studies and arriving at the final sample of studies that were included in the review. Specific comments related to the methods are as follows:

1. Line 112, authors should consider including more information on how the framework used was augmented and why there was a need to augment the framework

2. Lines 150, 142 and 189 seem to be conflicting in the range of the published studies used. While one set says 2000 – 2019, another says 2014 – 2019. Perhaps this is a typo and the authors meant 2014 – 2019 in the earlier lines.

3. Lines 158 – 164 is rather confusing and authors should consider improving the flow of information they are trying to convey.

4. Line 166 refers to umbrella reviews captured by this search. Did this manuscript utilize umbrella reviews or systematic reviews?

Results

The authors did a good job of providing summaries for the articles included in the review. Below are some comments regarding the result section of the manuscript

1. Appendix S1 is more informative than Figure 1, authors should consider which best serves their purpose and consider not using the other as it would be redundant.

2. Table 1, tables generally precludes numbering, the authors do not have to number the items on the inclusion/exclusion table and should consider removing the numbering within the rows and columns of this table

3. Appendix S6 does not contain the AMSTAR 2 that was utilized in the study and considering this, a thorough recheck of references may be helpful.

4. Figures are unlabeled. Authors should add labels to the figures within the manuscript.

Discussion

The discussion presents valuable information on the findings of the systematic literature review and highlighted limitations in the studies that were included in the umbrella review

1. I recommend that the authors consider using their subheadings to present the information in the discussion in a more structured format.

2. Authors should consider adding information on the impacts of the study and how the findings in their research can be applicable in preventing infectious diseases and health disparity among children LMIC’s.

3. The authors did not specify what infectious diseases they considered. Many infectious diseases exists, each with its unique method of transmission, infectivity and consequently different prevention strategies. The authors alluded to this in their statement on lines 228 – 229 and lines 725 - 727. Authors should consider narrowing their review to specific infectious diseases. Another option would be to group the infectious diseases according to related factors e.g. body systems (respiratory, intestinal, etc.) or based on types of vectors (mosquitoes, flies, etc.) or based on related preventive strategies. This may make their review more succinct and perhaps bring to light more strategies/interventions that are effective in reducing morbidity, mortality and health inequalities amongst children in LMIC’s.

Other Comments

1. Overall, the paper will benefit from a thorough rereading by a native English speaker to identify and address existing grammatical errors, long sentences, typos, missing words and improve the flow and readability of the text.

2. Redundant sentences exist in the manuscript for example line 486 Vs 505, the exclusion criteria is already on a table and needs not be spelled out in lines 125 – 128. I suggest that the authors consider using one form or the other and not both.

6. PLOS authors have the option to publish the peer review history of their article (what does this mean?). If published, this will include your full peer review and any attached files.

Reviewer #1: No

Reviewer #2: No

---

## [Author Response · Author response to Decision Letter 0]

24 Mar 2021

Dear Dr. Salinas-Miranda,

On behalf of our team, I would like to thank you for your editorial comments and for overseeing the review of our manuscript. Please convey our thanks to the reviewers for their detailed and constructive feedback. We have done our best to address these comments and make the requested clarifications and changes, as described below. The initial comments are numbered, followed by our reply, clarifications and, if relevant, quotes from the manuscript’s text where the modifications have been made. Please note that the lines and page numbers are based on the manuscript without tracked changes. We hope that these changes have helped making this manuscript clearer and more impactful. A number of changes in our manuscript are also the result of our updated search, which includes all relevant reviews until January 2021. 

Editorial requests:

The manuscript has been reformatted and the files renamed accordingly. If issues persist, please do let us know.

2. Please clarify whether bias and heterogeneity of studies were evaluated in this review.

The risk of bias of individual studies covered by the reviews included in this umbrella review was evaluated as part of the quality appraisal (see Quality appraisal p. 12 and S6 Appendix for details on the tool used). The heterogeneity of reviews was explored as part of the synthesis (see, Data synthesis pp. 12-13). As we did not undertake a meta-analysis, these evaluations were reported narratively in the results and in the discussion (see Strengths and limitations of the evidence base).

3. At this time, we ask that you please update the search period in your search strategy in order to include studies that have been published within the last 12 months.

The search has been updated to include studies up to January 2021. See p.7 lines 141 and 148-149:

“The following databases were searched from January 2000 until January 2021.”

“The records were later limited to records published from January 2014 until January 2021.”

As described in S5 Appendix, the search strings were run again for the period 2019-2021. However, four databases (Cochrane Library, Health Evidence, the Campbell Collaboration Library of Systematic Reviews and 3ie Evidence Hub) had significantly modified their search interfaces since our initial search. Thus, our search strings had to be revised to fit the new interface. In these cases, we run the search from 2000 onwards, as we had done for the initial search, to make sure that our search also captured any reviews that had been added or indexed differently has part of the databases’ update. Both search strings are described in S5 Appendix.

Nineteen additional reviews were included as part of this update, including 1 review from 2021, seven from 2020, eight from 2019 and four from before 2019 (resulting from re-running the whole search in the four databases which interface had been significantly modified). 

4. We note that Figure 4 in your submission contain map images which may be copyrighted. 

Thank you for sharing these links with us. We have remade the map using the Natural Earth files.

5. In your Data Availability statement, you have not specified where the minimal data set underlying the results described in your manuscript can be found. 

As described in our ethical approval section, this umbrella review exclusively relied on published reviews. A list and description of these reviews is available in S9 appendix, which we have also updated and uploaded as part of the resubmission.

6. Your ethics statement should only appear in the Methods section of your manuscript. 

The ethical approval information has been moved to the Method section, p.13, lines 244-246:

“This review exclusively worked with anonymous, group-level information available from published reviews. As a result, there is no risk to identifying individual data or disclosing confidential information. This study did not require ethical approval.”

Reviewer #1: 

1. Page 21 line 258 Quality of the evidence: because low and critically low quality studies were included in several analyses, the manuscript will be enhanced if at the end of each section the authors will include only high and moderate quality studies (resembling a sensitivity analysis). I see a summary in some of the sessions at the end of the manuscript but not at the beginning.

Thank you for pointing out this incoherence. We have added a synthesis of findings for moderate- to high-quality reviews in each section, except those that only included low- and critically low-quality reviews. See for example, p. 23, lines 382-385:

“Looking specifically at the two high-quality reviews,[62,64] both highlighted the heterogeneity and uneven quality of the included studies and their findings. This may partially explain the contradictory conclusions these two reviews drew on the effect of incentive-based programs.”

2. Page 33 line 420 Protecting children from infectious diseases. The authors did not find any review that protected specifically children from infectious diseases and approached population and health inequalities. I am wondering if the reviewers included all interventions guided to promote vaccination. Clarifying this point will be helpful.

We follow the World Health Organization Regional office for Europe’s definition of the three essential public health operations (see published protocol and manuscript p.6 lines 114-121). While this framework acknowledges the overlaps between the three operations, it categorizes vaccination intervention as prevention rather than protection. As a result, interventions aiming to promote vaccination have been included and analyzed under “Preventing infectious diseases in children” rather than “Protecting children from infectious diseases”.

Reviewer #2: 

Abstract

1. The authors stated in the study objectives that the focus of the umbrella review was for the purpose of “identify public health interventions that are effective in reducing mortality, morbidity and health inequalities from infectious diseases in low- and middle-income countries. This seems like a reach in terms of the findings and discussions of the paper.

We agree that the objective we set out for this review was ambitious. However, while we started with such a broad and ambitious objective, the findings are based on available evidence. We believe that the numerous gaps we found in the regard to this broad objective are important findings in their own rights. We have rephrased part of the conclusion to make this contrast clearer.

2. Line 28, should be more specific in line with the aim of the manuscript

The sentence has been rephrased as follows:

“It is imperative to understand the best available evidence concerning which public health interventions reduce morbidity, mortality and health inequalities in children aged under five years.”

3. Line 29 says to address these gaps, but only one gap was mentioned in the previous statement(s).

This sentence has been replaced to “To address this gap, we carried out an umbrella review (a systematic reviews of reviews) to identify evidence on the effects of public health interventions (promotion, protection, prevention) on morbidity, mortality and/or health inequalities due to infectious diseases amongst children in LMICs.”

Introduction

4. Lines 55 – 60. There is established evidence that education is related to income, but this relationship seems to be lost as the authors skip from talking about income to talking about education. A bridging sentence to explain the correlation between income and education will explain the cross from income to education.

We agree that the link between education and income is an important yet complex one when it comes to health inequalities. However, the sentence the reviewer refers to merely aims to illustrate the existence of child health inequalities according to various socio-economic factors for which income and education are examples. Exploring the complex link between the two factors and its implication for public health intervention is beyond the scope of this review. The lines in question have been rephrased and edited to clarify our point, p.3, Line 53-58

“Within countries, inequalities in health have been identified across many socio-economic factors. Household surveys from the Millennium Development Goals era show that children from the poorest households are almost twice as likely to die before the age of five compared to their wealthiest counterparts.[1] Similarly, children born from mothers with no education are almost three times as likely to die than those born from mothers with secondary or higher education.[1]”

5. Living in poor households, offspring of mothers with no education and inequalities for children living in rural areas are considered deprived groups, therefore I suggest that the “Inequalities are also found within deprived groups” found on line 60 be moved to line 56

These statements aim to reflect different definitions and understanding of inequalities, which we develop further in out model. The section has been rephrased to make this point clearer to the reader, p. 3, line 58-61:

“Inequalities are not only found between the most privileged and the most deprived groups but also within deprived groups. For example, a 2018 report by the United Nations Children's Fund (UNICEF) reveals that inequalities in child health indicators and outcomes not only vary between rural and urban populations but also within urban populations.[6]”

6. The statement on line 65 requires a citation.

A reference to the latest IHME Global Burden of diseases estimates have been repeated here. This statement is developed in the following sentences, p. 3-4 lines 62-66:

“Despite major progress in fighting infectious diseases,[7,8] the disease burden in children under five remains significant, especially in low- and middle-income countries (LMICs).[9] According to IHME 2019 GBD estimates, lower respiratory infections such as pneumonia, diarrheal diseases, malaria, HIV/AIDS and even vaccine-preventable diseases like measles remain among the leading causes of deaths and illness in children under five living in LMICs. [9]”

7. Lines 66 – 70 talks about leading causes of death for children in LMIC, I suggest that rather than have multiple sentences talking about the same issue, group the Pneumonia, diarrheal disease, Malaria, and HIV together in one sentence. The addition of the DALYs in those lines does not seem to add further information to the manuscript and I suggest it should be deleted.

This section has been rephrased and shortened accordingly, p. 3-4 lines 62-66 

“Despite major progress in fighting infectious diseases,[7,8] the disease burden in children under five remains significant, especially in low- and middle-income countries (LMICs).[9] According to IHME 2019 GBD estimates, lower respiratory infections such as pneumonia, diarrheal diseases, malaria, HIV/AIDS and even vaccine-preventable diseases like measles remain among the leading causes of deaths and illness in children under five living in LMICs. [9]”

8. Line 71, what diseases did the authors consider? The authors should be specific what types of infectious diseases they looked at.

We included all infectious diseases meeting the definition from the 11th International Classification of Diseases. This reference has been added in line 80 while the full definition is provided in our published protocol (doi:10.1136/bmjopen-2019-032981; S2 Appendix) “conditions caused by a pathogenic organism or microorganism, such as a bacterium, virus, parasite or fungus.”

“In this umbrella review, we aim to address some of these gaps by searching for public health interventions that are effective in reducing morbidity, mortality and health inequalities from infectious diseases (as defined by the 11th International Classification of Diseases[21])amongst children under five years of age living in LMICs.”

Methods

9. Line 112, authors should consider including more information on how the framework used was augmented and why there was a need to augment the framework

The changes we refer to in this sentence are the three core public health functions described immediately after. We’ve slightly rephrased this sentence to clarify that this change is to better take into account the public health intervention focus of our review, p. 6, line 111-115:

“To reflect this umbrella review’s focus on public health interventions, our framework was augmented with the three core public health functions or services identified by the World Health Organization Regional office for Europe’s (WHO EURO) as essential public health operations[30]: promoting child health, protecting children from infectious diseases and preventing such diseases.”

Additionally, a more detailed explanation of the framework is available in the published protocol (doi:10.1136/bmjopen-2019-032981, provided in S2 Appendix).

10. Lines 150, 142 and 189 seem to be conflicting in the range of the published studies used. While one set says 2000 – 2019, another says 2014 – 2019. Perhaps this is a typo and the authors meant 2014 – 2019 in the earlier lines.

We thank the reviewer for pointing out the mistake on line 170, which has been corrected accordingly. As the search has been updated as part of this revision, the dates covered by this review have been updated throughout the review:

p.7, Line 141: “The following databases were searched from January 2000 until January 2021”

p. 7, Line 148-149: “The records were later limited to records published from January 2014 until January 2021.”

11. Lines 158 – 164 is rather confusing and authors should consider improving the flow of information they are trying to convey.

These paragraphs have been rephrased for clarity, p. 8 lines 155-163 

“In addition to the searching of these databases, we (EB) performed a manual search in Google Scholar and on the following international organizations’ websites, using selected keywords: 

- UNICEF Office of Research – Innocenti [37] 

- UNICEF [38,39] 

- World Health Organization (WHO) [40,41] 

The full search strategy for both the databases and manual searches can be found in S5 Appendix.”

12. Line 166 refers to umbrella reviews captured by this search. Did this manuscript utilize umbrella reviews or systematic reviews?

Our umbrella reviews only included systematic reviews, as detailed in our inclusion criteria. However, to make sure that our search didn’t miss any important reviews, we also manually searched the reference list of relevant umbrella reviews captured by our search. 

These elements have been clarified in p. 8 Lines 164 to 167 

“Finally, and in addition to what was stated in our protocol, we (EB/TM or EB/DS) conducted citation follow-up by searching the reference lists of umbrella reviews captured by this search for potential additional records. The umbrella reviews themselves were excluded from our review.”

And in Table 1, under exclusion criteria:

“The publication is a primary study, an umbrella review, a conference proceeding or paper, an abstract, editorial, letter, comment, erratum, survey, note or a doctoral thesis; or does not meet one or more of the three key elements of systematic reviews and evidence syntheses as defined in PICOS; or does not synthesize at least 2 relevant primary studies.”

Results

13. Appendix S1 is more informative than Figure 1, authors should consider which best serves their purpose and consider not using the other as it would be redundant.

The two figures are complementary. Figure 1 presents our main theoretical framework (used to guide our search, extraction and synthesis). It includes the intervention’s characteristics (level and public health functions), equity characteristics (approach and objective) and its effects. Appendix S1 is an illustration of the first element, namely the intervention’s characteristics. It was developed further to early comments on our protocol, to help make selected elements of our framework more concrete to the reader. The table in Appendix S1 was part of the protocol alongside Figure 1 but has been moved to the appendices here in an attempt to keep the method section short yet clear. Hence, we believe that removing one or the other could be detrimental to the reader’s understanding of our method and approach. 

14. Table 1, tables generally precludes numbering, the authors do not have to number the items on the inclusion/exclusion table and should consider removing the numbering within the rows and columns of this table

The numbers have been removed from Table 1, as suggested by the reviewer. 

15. Appendix S6 does not contain the AMSTAR 2 that was utilized in the study and considering this, a thorough recheck of references may be helpful.

We reformatted the AMSTAR checklist’s results in the table in pp. 4-5 in Appendix S6 to make it easier to compile and compare the weaknesses and quality of all the reviews included in our umbrella review. However, the reviewers did apply the AMSTAR checklist as available on the AMSTAR website, alongside the AMSTAR guidance document, for each review, and then reported their answer for each question in the table found in Appendix S6. The reference found in the Appendix and the main text is the reference provided by the AMSTAR website’s checklist page. 

We have added the guidance described here to Appendix S6 to clarify this point, p.4:

“When assessing individual studies, please refer to the AMSTAR 2 checklist and guidance document. (Shea et al., 2017) Then, please report your answer in the table below.”

16. Figures are unlabeled. Authors should add labels to the figures within the manuscript.

We apologize for this issue as we had misunderstood how the figures would be uploaded. We have added the titles and legend to individual figure file. We hope that this has solved the issue. 

Discussion

17. I recommend that the authors consider using their subheadings to present the information in the discussion in a more structured format.

The discussion does not follow the same structure as the results in order to better highlight the strength of the evidence and whether effective interventions had been identified to address the burden of infectious diseases at population level, on health inequalities or both. After exploring various options, we had agreed that this option was clearer, shorter and easier to read than mirroring the structure of the results. 

As current subsections are fairly short already, we believe that adding further sub-headings would be rather cumbersome and may distract the reader from the key messages. We have revised some of the existing sub-headings and added a subsection on the review’s implication for practice, to help make the Discussion clearer to the reader. 

18. Authors should consider adding information on the impacts of the study and how the findings in their research can be applicable in preventing infectious diseases and health disparity among children LMIC’s.

We added a sub-section on the review’s Implication for research, policy and practice in the discussion (pp.52-54).

19. The authors did not specify what infectious diseases they considered. Many infectious diseases exists, each with its unique method of transmission, infectivity and consequently different prevention strategies. The authors alluded to this in their statement on lines 228 – 229 and lines 725 - 727. Authors should consider narrowing their review to specific infectious diseases. Another option would be to group the infectious diseases according to related factors e.g. body systems (respiratory, intestinal, etc.) or based on types of vectors (mosquitoes, flies, etc.) or based on related preventive strategies. This may make their review more succinct and perhaps bring to light more strategies/interventions that are effective in reducing morbidity, mortality and health inequalities amongst children in LMIC’s.

As further clarified in the method, we included all infectious diseases meeting the definition from the 11th International Classification of Diseases. We decided on such a broad scope not only as a way to provide a good overview of the field but also, to identify and highlight interventions that could affect the burden of several diseases at once. As our summary Table 3a and 3c show, interventions such as nutrition supplement or water, sanitation and hygiene interventions have the potential to address several diseases at once. We believe that such findings are important contributions to decision-making in resource-constrained settings facing competing health priorities. However, we fully agree with the reviewer that diseases’ specificities, modes of transmission, burden of diseases or public health systems are essential considerations when interpreting and using the findings of this review. Therefore, we have added a paragraph on these consideration in the discussion, as part of the new Implication for research, policy and practice section (pp. 52-54, lines 932-942):

“It should be noted again that this review was focused on specific outcomes affecting morbidity, mortality and health inequalities from infectious diseases amongst children in LMICs. Therefore, the findings presented here should be assessed and understood within the broader body of public health evidence addressing the needs of children but also other populations or affecting other causes of ill-health. Additionally, the findings of this review should be interpreted within the specific burden of infectious diseases faced by an area, country or region. To provide this broad overview of the field and identify interventions that can offer health gains on multiple diseases outcomes, we purposely refrained from a disease- or transmission mode-specific approach. Yet, these are essential components in the choice of intervention, its design and its chances of success. Hence, the findings of this review should be carefully interpreted according to the specific context and public health system in which they are to be used.”

Other Comments

20. Overall, the paper will benefit from a thorough rereading by a native English speaker to identify and address existing grammatical errors, long sentences, typos, missing words and improve the flow and readability of the text.

The native English-speaking authors of this paper have proof-read the revised manuscript. 

21. Redundant sentences exist in the manuscript for example line 486 Vs 505, the exclusion criteria is already on a table and needs not be spelled out in lines 125 – 128. I suggest that the authors consider using one form or the other and not both.

We undertook a thorough review of the text as part of our revision and update to remove redundancies and errors. However, some of the redundant information is present by design, especially in regard to tables. For example, Tables 1 and 3a to 3c aim to provide a summary of key information while the text provides further details and references to complement this information. 

Thank you for your time, help and advice. We look forward to hearing your assessment on the revised manuscript. 

Sincerely

---

## [Decision Letter · Decision Letter 1]

6 May 2021

Which public health interventions are effective in reducing morbidity, mortality and health inequalities from infectious diseases amongst children in low- and middle-income countries (LMICs): an umbrella review.

PONE-D-20-17157R1

Dear Dr. Besnier,

We’re pleased to inform you that your manuscript has been judged scientifically suitable for publication and will be formally accepted for publication once it meets all outstanding technical requirements.

Kind regards,

Abraham Salinas-Miranda, MD, PhD

Academic Editor

PLOS ONE

Additional Editor Comments (optional):

Thank you for addressing all requested revisions. The decision is to accept for publication.

Journal Requirement:

1. Please clarify in the Methods section the motivations for the chosen start dates for the search, for both 2000 and 2014.

Reviewers' comments:

Reviewer's Responses to Questions

**Comments to the Author**

1. If the authors have adequately addressed your comments raised in a previous round of review and you feel that this manuscript is now acceptable for publication, you may indicate that here to bypass the “Comments to the Author” section, enter your conflict of interest statement in the “Confidential to Editor” section, and submit your "Accept" recommendation.

Reviewer #1: All comments have been addressed

Reviewer #2: All comments have been addressed

2. Is the manuscript technically sound, and do the data support the conclusions?

Reviewer #1: Yes

Reviewer #2: Yes

3. Has the statistical analysis been performed appropriately and rigorously? 

Reviewer #1: Yes

Reviewer #2: Yes

4. Have the authors made all data underlying the findings in their manuscript fully available?

Reviewer #1: Yes

Reviewer #2: Yes

5. Is the manuscript presented in an intelligible fashion and written in standard English?

Reviewer #1: Yes

Reviewer #2: Yes

6. Review Comments to the Author

Reviewer #1: I think this is a complete review. The authors did a very good job answering all questions the reviewers have posted in the initial review. I particularly agree with the distribution of the tables and figures. I think the manuscript flows better after the reviewers questions and comments were addressed.

Reviewer #2: (No Response)

7. PLOS authors have the option to publish the peer review history of their article (what does this mean?). If published, this will include your full peer review and any attached files.

Reviewer #1: No

Reviewer #2: No

---

## [Editor Report · Acceptance letter]

21 May 2021

PONE-D-20-17157R1 

Which public health interventions are effective in reducing morbidity, mortality and health inequalities from infectious diseases amongst children in low- and middle-income countries (LMICs): an umbrella review. 

Dear Dr. Besnier:

I'm pleased to inform you that your manuscript has been deemed suitable for publication in PLOS ONE. Congratulations! Your manuscript is now with our production department. 

Kind regards, 

on behalf of

Dr. Abraham Salinas-Miranda 

Academic Editor

PLOS ONE